# A synaptomic analysis reveals dopamine hub synapses in the mouse striatum

Vincent Paget-Blanc [1,7], Marlene E. Pfeffer [1,7], Marie Pronot [1,7], Paul Lapios[1], Maria-Florencia Angelo [1], Roman Walle[2], Fabrice P. Cordelières [3], Florian Levet[1,3], Stéphane Claverol [4], Sabrina Lacomme [3], Mélina Petrel [3], Christelle Martin[1], Vincent Pitard[5], Véronique De Smedt Peyrusse[2], Thomas Biederer [6], David Perrais [1], Pierre Trifilieff [2] & Etienne Herzog [1✉]

Dopamine transmission is involved in reward processing and motor control, and its impairment plays a central role in numerous neurological disorders. Despite its strong pathophysiological relevance, the molecular and structural organization of the dopaminergic synapse remains to be established. Here, we used targeted labelling and fluorescence activated sorting to purify striatal dopaminergic synaptosomes. We provide the proteome of dopaminergic synapses with 57 proteins specifically enriched. Beyond canonical markers of dopamine neurotransmission such as dopamine biosynthetic enzymes and cognate receptors, we validated 6 proteins not previously described as enriched. Moreover, our data reveal the adhesion of dopaminergic synapses to glutamatergic, GABAergic or cholinergic synapses in structures we named "dopamine hub synapses". At glutamatergic synapses, pre- and postsynaptic markers are significantly increased upon association with dopamine synapses. Dopamine hub synapses may thus support local dopaminergic signalling, complementing volume transmission thought to be the major mechanism by which monoamines modulate network activity.

[1] Univ. Bordeaux, CNRS, Interdisciplinary Institute for Neuroscience, IINS, UMR 5297, F-33000 Bordeaux, France. [2] Univ. Bordeaux, INRAE, Bordeaux INP, NutriNeuro, UMR 1286, F-33000 Bordeaux, France. [3] Univ. Bordeaux, CNRS, INSERM, Bordeaux Imaging Center, BIC, UAR 3420, US 4, F-33000 Bordeaux, France. [4] Univ. Bordeaux, Plateforme Proteome, 33000 Bordeaux, France. [5] UB'FACSility CNRS UMS 3427, INSERM US 005, Univ. Bordeaux, F-33000 Bordeaux, France. [6] Department of Neurology, Yale School of Medicine, New Haven, CT 06511, USA. [7] These authors contributed equally: Vincent Paget-Blanc, Marlene E. Pfeffer, Marie Pronot. ✉email: etienne.herzog@u-bordeaux.fr

Since the 1950s with the first ultrastructural characterization of the synapse in the central nervous system[1] a wide variety of synapse types has been described based on morphological criteria[2]. The archetypal synapse type is the so-called asymmetric excitatory synapse on dendritic spines[1], which represent the vast majority of synapses (~80%). Its ultrastructure is easily identifiable in the tissue by the presence of a postsynaptic density and a dense cluster of synaptic vesicles, and it has been extensively studied in vitro using primary neuronal cultures[3]. Alternatively, symmetric synapses are predominantly inhibitory or modulatory. They do not display postsynaptic densities and are more difficult to identify in situ[4,5]. Moreover, many types of synaptic organizations are not abundant enough and/or accessible with in vitro models. These limitations hinder our understanding of neuronal network functioning.

While glutamate and GABA (Gamma-Amino Butyric Acid) neurotransmissions drive point to point information locally, modulatory neurotransmitters pace regional activity through volume transmission in the neuropil[6,7]. Dopamine transmission is a major neuro-modulatory system involved in several functions such as movement initiation, reward prediction error and incentive processes, notably by its projections onto spiny projection neurons (SPNs) of the striatum[8]. Dopamine signalling is presumed to modulate glutamate transmission onto SPNs through the release of dopamine mainly from varicosities devoid of synaptic differentiation. Previous investigations assessing the presence of dopaminergic synapses found that only a minority of axon terminals form synapses onto SPN spines, dendrites, or presynapses[4,5,9]. Recent work also challenges the model of volume dopamine transmission by providing evidence for local point-to-point signalling. In particular, optophysiology approaches revealed rapid and local transmission at dopaminergic projections to the striatum[10–12], which is in accordance with the existence of the machinery allowing fast dopamine release at striatal varicosities[13]. Moreover, the distribution of varicosities in the striatal neuropil appears biased toward proximity with the surrounding glutamatergic synapses[5], and dopamine receptors interact physically and functionally with glutamate and GABA receptors[14–17], suggesting a tight coupling between dopamine, glutamate and/or GABA signalling.

In the present work, we unravel the cellular and molecular synaptome of a single projection pathway[18]. This critically complements current connectomic approaches using optophysiology and tracing methods, which are limited in terms of molecular analysis of specific synapses at play in a given circuit[19]. To that end, we established a workflow combining fluorescence tracing of the dopaminergic pathway, fluorescence-activated synaptosome sorting[20,21] and an array of semi-quantitative analysis methods ranging from conventional immunofluorescence characterization to mass spectrometry-based proteomics. With this approach, we provide a proteome and validate 6 new proteins (Cpne7, Apba1/Mint-1, Cadps2, Cadm2/SynCAM 2, Stx4, Mgll) enriched at dopaminergic synapses from the mouse striatum. Moreover, we show a physical coupling between dopaminergic and other synapses in a tight multipartite complex that we name "dopamine hub synapses".

## Results

### Fluorescence-activated synaptosome sorting (FASS) of dopaminergic synaptosomes reveals multipartite synaptic hub structures.

We labelled the dopaminergic projection onto the striatum through stereotaxic injection of an adeno-associated viral vector carrying Cre-dependent EGFP[22] in the midbrain of Dopamine Transporter promoter (DAT)-Cre transgenic mice[23] (Fig. 1a). We applied a classical synaptosome fractionation miniaturized to 1.5 ml tubes as previously published[24,25] to generate samples labelled with EGFP (Fig. 1a). To validate our labelling and fractionation approach, we performed a complete subcellular fractionation of the dissected striata and measured the amount of two soluble reporter proteins, tyrosine hydroxylase (Th), that catalyses the limiting step for dopamine synthesis[26], and the fluorescent reporter EGFP. They were probed using a semi-automatic capillary immunoblot system producing electrophoregrams (Fig. 1b) or membrane-like band patterns (Fig. 1c). Quality controls of the fractionation show the enrichment of synaptophysin (Syp) in synaptosomes (SYN) and crude synaptic vesicle (LP2) fractions, while the plasma membrane glutamate transporter GLAST (Slc1a3/GLAST) is enriched in synaptic plasma membranes (SPM). We confirm the high concentration of Th and EGFP signals in synaptosomes (SYN) and derived soluble fractions (LS1 and LS2) while they are weak in nuclear P1 and cytosolic S2 fractions, relative to homogenate (H) (H: Th = 1 ± 0.26, EGFP = 1 ± 0.53; P1: Th = 0.36 ± 0.07, EGFP = 0.23 ± 0.09; S2: Th = 0.77 ± 0.1, EGFP = 0.49 ± 0.23; P2: Th = 1.2 ± 0.27, EGFP = 1.12 ± 0.52; SYN: Th = 1.5 ± 0.31, EGFP = 1.48 ± 0.71; LS1: Th = 2.2 ± 0.19, EGFP = 1.8 ± 0.5; SPM: Th = 0.26 ± 0.02, EGFP = 0.07 ± 0.02; LS2: Th = 1.58 ± 0.32, EGFP = 1.29 ± 0.22; LP2: Th = 0.82 ± 0.4, EGFP = 0.03 ± 0.01; N = 3 complete fractionations; Fig. 1c, d). Based on these reporters, we can conclude that most of the cytosolic content of dopaminergic axons is present in the synaptosome fraction (SYN) and available for discrimination by the fluorescence-activated synaptosome sorting (FASS)[21,27] procedure (Fig. 1a). Of note, a small fraction of Th seems associated with light membranes of the crude synaptic vesicle fraction, an observation reminiscent of the one made with GABA synthesizing enzymes[28].

FASS[21,27] applied to this sample allowed recovering up to 35 million fluorescent synaptosomes according to cytometer counts (Fig. 1a). The collection of an equal number of singlet particles regardless of their EGFP fluorescence status served as a control for all comparisons (SYN). Bulk FASS samples were used to perform immunoblot and mass spectrometry assays. In addition, we established the immobilization of particles on glass coverslips to analyze them through quantitative immunofluorescence, super-resolution STED microscopy and electron microscopy (Fig. 1a). Our gating strategy was adapted from the previous work[27] to avoid sorting aggregated particles, i.e. particles with high forward scatter (FSC) and side scatter (SSC) values and sort specifically singlets, particles with FSC values around 0 (Supplementary Fig. 1a–c). Among singlets, EGFP+ events are specifically detected by setting a fluorescence threshold from the autofluorescence of synaptosomes from non-injected mice (Supplementary Fig. 1b). Synaptosomes from DAT-Cre-EGFP mice (SYN) contained on average 3.86 ± 0.53% EGFP+ synaptosomes (N = 9 sorts; Fig. 1e, f). Upon reanalysis of the DA-FASS sample in the cell sorter, EGFP+ events represented around 50% of the total (48.9 ± 2.3%, N = 8 sorts; Fig. 1f) and EGFP− synaptosomes were concomitantly depleted (SYN: EGFP− Singlets = 66.1 ± 4%, N = 9 sorts; DA-FASS: EGFP− Singlets = 30.9 ± 2.8%, SYN-DA-FASS: Šídák's multiple comparison ****p < 0.0001 N = 8 sorts; Fig. 1f). Based on these values, we can expect an enrichment of a specific component of dopaminergic synaptosomes of 48.9/3.9 = 12.5-fold in DA-FASS relative to SYN samples. With the use of mNeonGreen as a fluorescence reporter we increased the yield up to 35 million mNG+ synaptosomes, which we compared to an equal number of singlet particles, regardless of their green fluorescence status (SYN; Supplementary Fig. 1e–g).

We further validated these sorts using capillary electrophoresis-based immunoblots. As expected, Th and the dopamine transporter (Slc6a3/DAT) display a strong enrichment

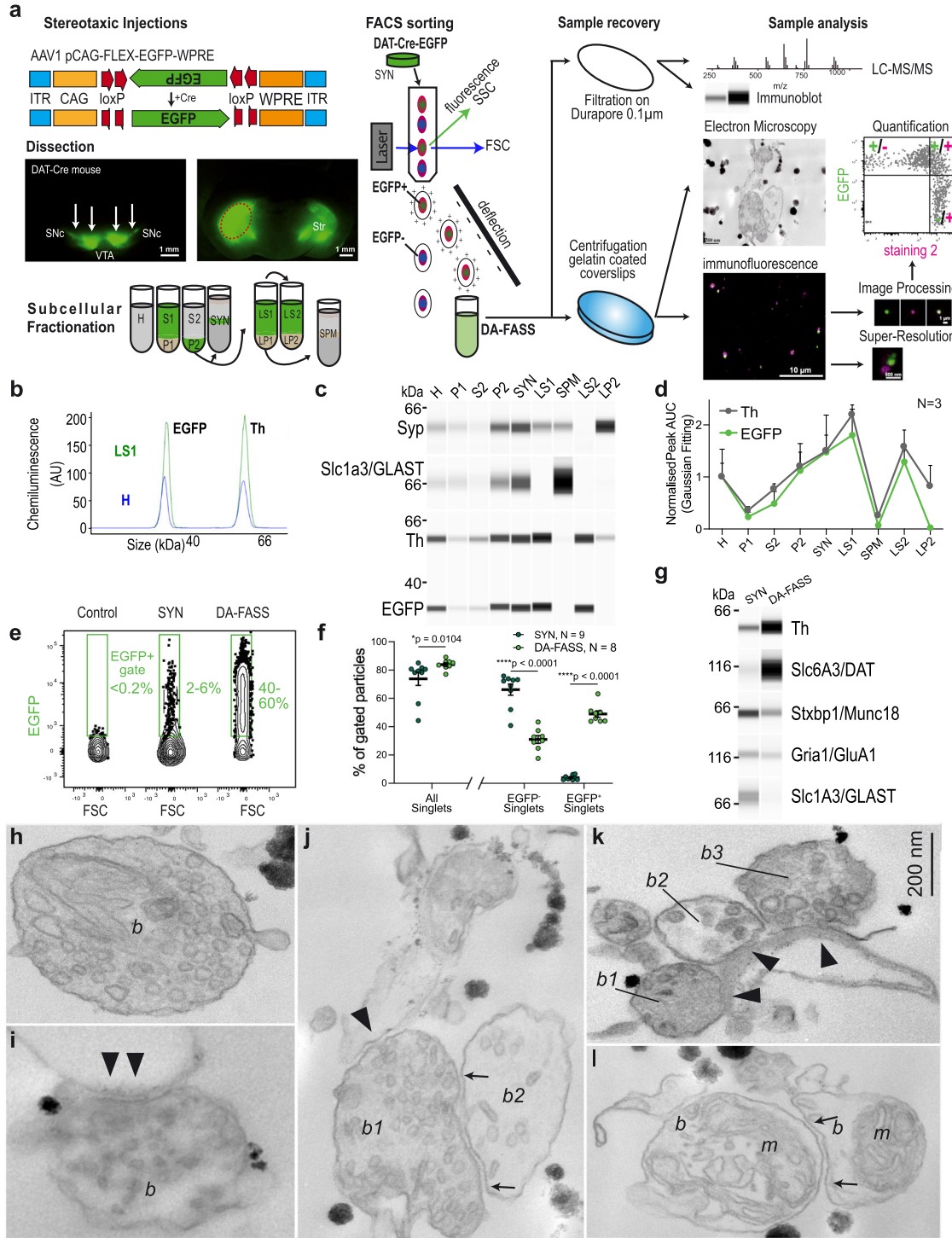

after DA-FASS. In contrast, GLAST is strongly depleted while the glutamate receptor (Gria1/GluA1) or the synaptic active zone protein Munc18 (Stxbp1) are reduced (Fig. 1g). We then performed qualitative ultrastructural analysis on DA-FASS samples using transmitted electron microscopy (TEM) (Fig. 1h–l). We identified synaptosome profiles with resealed presynaptic elements (Fig. 1h) and in some cases a clear adhesion with a postsynaptic membrane (Fig. 1i). Surprisingly, we also identified profiles displaying several presynapses organized around possible postsynaptic membranes (Fig. 1j–l). Most of the synaptosomes were cut at an angle that prevented the clear identification of all

synaptic elements (Fig. 1l). In another example, we found two distinct presynapses, one electron-dense terminal with a synaptic vesicle cluster adhering to a presynaptic element with few vesicles and to another compartment that could be dendritic (Fig. 1j). Finally, a postsynaptic element displayed adhesion to three different "boutons", one of them displaying a clearer background and fewer vesicles (Fig. 1k). Contrarily to aggregates, these multipartite synapses were preserved even though our procedure exposed them to shearing forces twice, first during tissue homogenization and second, in flight through the nozzle of the cell sorter[21,27,29] (see the workflow in Fig. 1a; Supplemental Figs. 1 and 5). Beyond displaying

**Fig. 1 Single projection fluorescence-activated synaptosome sorting (FASS) isolates dopaminergic hub synaptosomes. a** Workflow of DAT-cre/AAV-EGFP synaptosome sorting and analysis. DAT-Cre mice injected with a Cre-dependent AAV expressing EGFP or mNeonGreen (Supplementary Fig. 1) in the Substantia Nigra pars compacta and the Ventral Tegmental Area. Fluorescence guided dissection in the Striatum (Str, red-dashed circle). Subcellular fractionation and FASS. Collection on filters or glass coverslips for FASS sample analysis by mass spectrometry, immunoblot, electron microscopy, or immunofluorescence. **b–d** Analysis of subcellular fractionation through capillary electrophoresis immunoblot. **b** EGFP and Th chemiluminescence peaks for H (blue) and LS1 (green) fractions. **c** Chemiluminescence bands of Synaptophysin1, Slc1a3/GLAST, Th and EGFP. **d** Relative integrated intensity for Th (grey) and EGFP (green) for each subcellular fraction (H to LP2), (mean ± SEM from 3 independent fractionations; Two-way ANOVA: Interaction $F_{8,36} = 0.273$ $p = 0.971$, Fraction $F_{8,36} = 7.062$ ****$p < 0.0001$, Protein $F_{1,36} = 2.387$ $p = 0.131$). **e** Flow cytometry analysis of synaptosomes through DA-FASS sorting. The EGFP + gate was set to have 0–0.2% of events in control SYN samples. Before sorting SYN show 2–6% of EGFP + synaptosomes while DA-FASS contains 40–60% of EGFP + synaptosomes. **f** Averaged percentage of gated particles from SYN ($N = 9$) and DA-FASS ($N = 8$) biologically independent synaptosome preparations (mean ± SEM; Two-tailed Mann–Whitney test for all singlets SYN-DA-FASS *$p = 0.01$ and Two-way ANOVA for EGFP ± singlets: Interaction $F_{1,30} = 218.3$ ****$p < 0.0001$, Gating $F_{1,30} = 66.42$ ****$p < 0.0001$, Condition $F_{1,30} = 3.302$ $P = 0.079$ with Šídák's multiple comparisons test). **g** Immunoblot against Th, DAT, Munc18, GluA1 and GLAST through DA-FASS ($N = 1$). **h–l** Electron micrographs of sorted synaptosomes ($N = 2$). **h, i** Typical synaptosomes displaying a bouton (b), synaptic vesicles (SV) and an opened postsynaptic membrane (arrowheads in **i**). **j** Example of a multipartite synaptosome displaying a SV-rich bouton (b1) contacting a postsynaptic membrane (arrowheads) and a second bouton (arrows) less populated with SVs (b2). **k** Multipartite synaptosome displaying 3 distinct presynaptic profiles (b1, b2, and b3) contacting a postsynaptic membrane (arrowheads). **l** Multipartite synaptosome cut through a plane that is not optimal. Boutons (b), Mitochondria (m). Scale bar, 200 nm; for uncropped immunoblots see supplementary material or original data files in source data.

axo-axonic or axo-spinous synapses, our fractionation isolates multipartite bound synaptic elements that we name "dopamine hub synapses".

**DA-FASS synaptosomes display pre- and postsynaptic features of dopaminergic synapses.** To further characterize the dopaminergic hub synapses, we compared SYN synaptosomes with DA-FASS synaptosomes immobilized on coverslips and immunolabelled with dopaminergic markers. Individual synaptosomes were quantified according to EGFP and dopaminergic markers intensity. Quadrant gates were defined to split positives and negatives for each label (Fig. 1a). The top 2 quadrants are EGFP$^+$ synaptosomes and percentages of particles are displayed in each quadrant. Th$^+$/EGFP$^+$ synaptosomes population rose from 57% of the total population before sort to 83% after sort (SYN: EGFP$^+$/Th$^+$ = 57.4 ± 2.8%, $N = 2$ sorts, $n = 13$ fields of view; DA-FASS: EGFP$^+$/Th$^+$ = 83.1 ± 1.1%, $N = 3$ sorts, $n = 21$ fields of view; SYN-DA-FASS Šídák's multiple comparison ****$p < 0.0001$; Fig. 2a, b). Stimulated Emission Depletion (STED) imaging, which has a resolution of 30 nm, below synaptosome size of about 500 nm (Fig. 1h–l) revealed that Th signals were highly co-localized with EGFP (Fig. 2d). Similarly, we found a strong co-localization of EGFP$^+$ synaptosomes with DAT (Dopamine Transporter) signal (SYN: EGFP$^+$/DAT$^+$ = 14.5%; DA-FASS: EGFP$^+$/DAT$^+$ = 47%; $N = 1$ sort; Supplementary Fig. 2a, b). As expected from the immunoblot analysis (Fig. 1g), the marker Slc1a3/GLAST that labels astrocytic membranes was not significantly associated with the EGFP$^+$ synaptosomes (SYN: EGFP$^+$/GLAST$^+$ = 1%; DA-FASS: EGFP$^+$/GLAST$^+$ = 6%; $N = 1$ sort; Supplementary Fig. 2c, d). These data further confirm that EGFP$^+$ synaptosomes bear genuine dopaminergic synaptic markers and are strongly enriched through DA-FASS.

We then explored the co-segregation of dopamine receptors type 1 and -2 (D1R, D2R) together with EGFP$^+$ synaptosomes. D1R co-enriched almost 10-fold (29/3) with DA-FASS (SYN: EGFP$^+$/D1R$^+$ = 3.2 ± 0.7%, $N = 2$ sorts, $n = 12$ fields of view; DA-FASS: EGFP$^+$/D1R$^+$ = 28.6 ± 2%, $N = 3$ sorts, $n = 22$ fields of view; SYN-DA-FASS Šídák's multiple comparison ****$p < 0.0001$; Fig. 2e–g), while D1R$^+$/EGFP$^-$ events were depleted two-fold (SYN: EGFP$^-$/D1R$^+$ = 86.8 ± 1.4%, $N = 2$ sorts, $n = 12$ fields of view; DA-FASS: EGFP$^-$/D1R$^+$ = 43.2 ± 2%, $N = 3$ sorts, $n = 22$ fields of view; SYN-DA-FASS Šídák's multiple comparison ****$p < 0.0001$; Fig. 2e–g bottom right quadrants). Fifty one percent of EGFP$^+$ synaptosomes (29/(29 + 28) × 100) were labelled for D1R (DA-FASS: EGFP

$^+$/D1R$^-$ = 28.1 ± 1.4%, $N = 3$ sorts, $n = 22$ fields of view; Fig. 2e–g upper quadrants). D1R immunolabelling appeared as patches of staining apposed to EGFP$^+$ particles (Fig. 2h). D2R labels were found on more than 78% (53/(53 + 15) × 100) of EGFP$^+$ synaptosomes and co-enriched with EGFP (DA-FASS: EGFP$^+$/D2R$^+$ = 53.2 ± 2.3%, EGFP$^+$/D2R$^-$ = 14.6 ± 1.4%, $N = 3$ sorts, $n = 30$ fields of view; Fig. 2i–k upper quadrants). EGFP$^-$/D2R$^+$ events were depleted more than two-fold over DA-FASS (SYN: EGFP$^-$/D2R$^+$ = 72.2 ± 3%, $N = 3$ sorts, $n = 32$ fields of view; DA-FASS: EGFP$^-$/D2R$^+$ = 32.3 ± 2.4%, $N = 3$ sorts, $n = 30$ fields of view; SYN-DA-FASS Šídák's multiple comparison ****$p < 0.0001$; Fig. 2i–k lower right quadrants). With STED microscopy we detected D2R either co-localized with EGFP (putative autoreceptors), or distributed in patches apposed to EGFP (putative heteroreceptors, Fig. 2l) as described with immunogold electron microscopy[30], even though the exact nature of each patch cannot be readily established. Altogether, our data support the view that dopaminergic synaptosomes bear a postsynaptic element containing cognate receptors. Consistent with the fact that roughly half of SPNs express D1R and the other half D2R[31], 51% of EGFP$^+$ synaptosomes are associated with D1R expressing SPNs and the rest most likely with D2R, but our observation is confounded by the presence of D2R autoreceptors at both types of synaptosomes.

**Label-free semi-quantitative proteomics reveals 57 proteins highly enriched at DA-FASS synaptosomes.** To identify the molecular nature of dopaminergic synaptosomes, we generated a set of six DA-FASS samples and processed them for label-free quantification of proteins through mass spectrometry (MS). We accumulated 35 million mNeonGreen$^+$ synaptosomes from three independent DA-FASS experiments. All SYN singlets representing the conventional synaptosome preparation were used as control samples (3 in total). A total of 3824 proteins were identified with one peptide or more, throughout the six samples. Among these, 2653 proteins were identified robustly and quantified with at least 2 distinct peptides. We considered a significant difference between samples for proteins displaying a ratio greater than 1.5 in DA-FASS samples compared to SYN, with an adjusted $p$-value smaller than 0.05. Based on these criteria, 63 proteins are significantly depleted upon sorting while 57 others appear significantly enriched (Fig. 3g, Supplementary Table 3). The depleted proteins did not have a clear gene ontology signature and

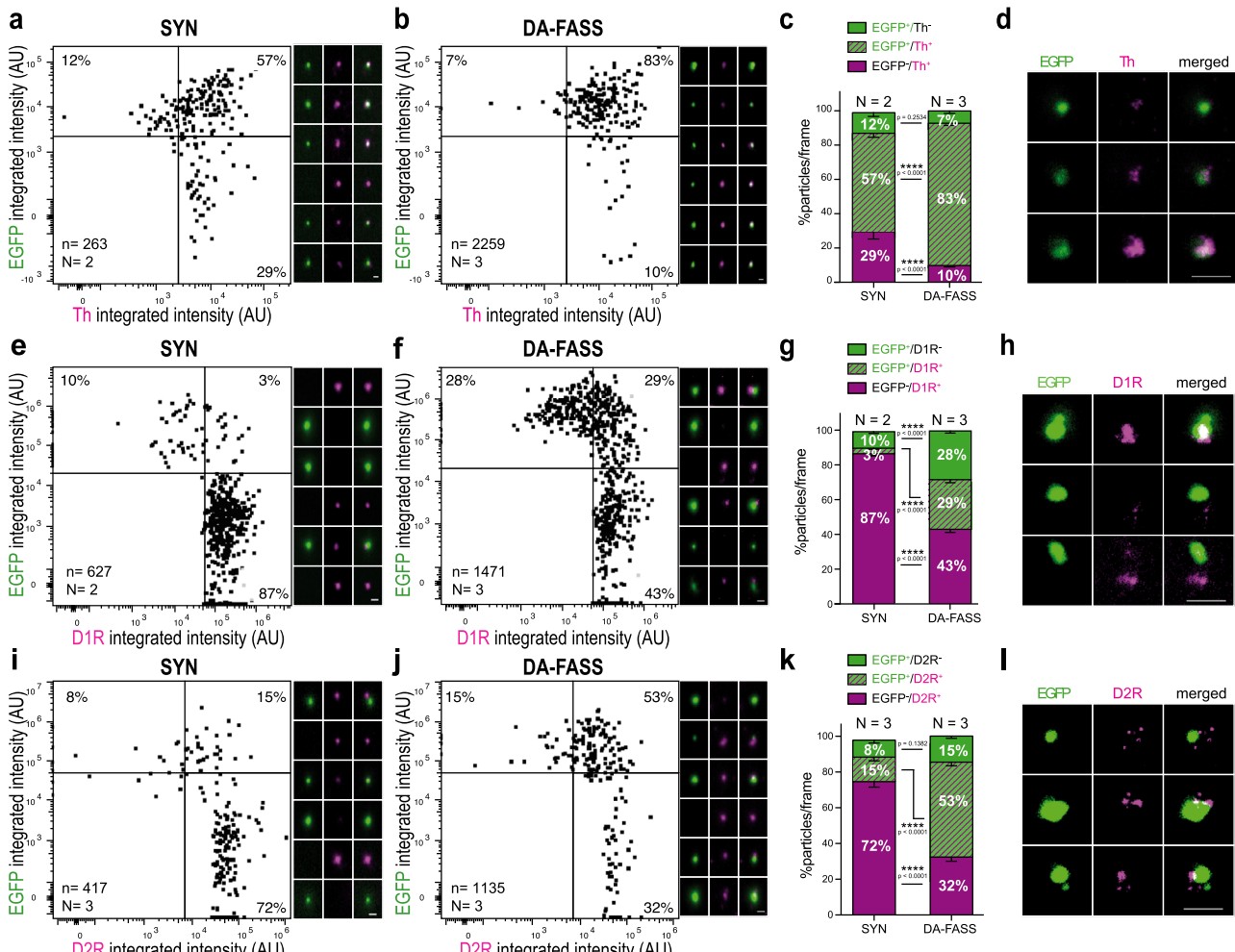

**Fig. 2 Immunofluorescence analysis of DA-FASS synaptosomes reveal the enrichment for pre- and postsynaptic dopaminergic markers.**
**a**, **b** Epifluorescence images of SYN and DA-FASS synaptosomes immobilized on coverslips and immunolabelled with anti-Th and anti-EGFP. Dot plot population analysis of fluorescence intensities in both channels. **c** Analysis of staining as in **a** and **b** showing particle proportions per frame. **d** STED images of EGFP (green) and Th (magenta) labelled synaptosomes. **e**, **f** Same as **a**, **b** for EGFP and D1 dopamine receptors. **g** Proportion of differently stained particles per frame. **h** STED microscopy detects D1 receptor clusters (magenta) apposed to the EGFP+ synaptosomes (green). **i**, **j** Same as a-b for anti-EGFP and anti-D2 dopamine receptors. **k** Proportion of differently stained particles per frame. **l** STED images display D2R (magenta) patches apposed to EGFP (green). All data are mean ± SEM and pulled from $N = 2$ to $N = 3$ independent sorts and $n = 4$ to $n = 11$ field of view per independent sort. Each independent sort pooled at least 3 animals. Statistical significance was analyzed using Two-way ANOVA; c EGFP/Th: Interaction $F_{2,96} = 65.04$ ****$p < 0.0001$, Condition $F_{1,96} = 0.034$ $p = 0.855$, Immunolabelling $F_{2,96} = 510.3$ ****$p < 0.0001$; **g** EGFP/D1R Interaction $F_{2,96} = 208$ ****$p < 0.0001$, Condition $F_{1,96} = 0.007$ $p = 0.931$, Immunolabelling $F_{2,96} = 437.9$ ****$p < 0.0001$; **k** EGFP/D2R: Interaction $F_{2,180} = 149.4$ ****$p < 0.0001$, Condition $F_{1,180} = 0.671$ $p = 0.414$, Immunolabelling $F_{2,180} = 163.5$ ****$p < 0.0001$ with Šídák's multiple comparisons test. For all panels, scale bar = 1 μm. See extra immunofluorescence analysis in Supplementary Fig. 2.

were not studied further. We thus focused on the enriched proteins.

We first compared the 2653 proteins dataset to the broad survey of mouse brain proteins produced by Sharma and colleagues[32]. Ninety percent of our dataset is common to the global mouse brain proteome. Among 158 proteins significantly enriched in the bulk dissection of the striatum, 89 are represented in the synaptosome samples which is consistent with the selectivity of our subcellular fractionation (Fig. 3b, c, Supplementary Table 4). 403 proteins are common with those previously identified to be specific to a given cell type. A heatmap analysis of these shows the main neuronal origin of our synaptosome samples (Fig. 3d, Supplementary Table 4, Supplementary Fig. 3). We then compared the obtained proteome with the curated database of known synaptic gene ontologies (SynGO)[33]. Among 2653 genes from this proteome, 684 genes are documented in

SynGO covering all localizations and functions reported in the second level of SynGO terms. This gene set is associated with synapse organization (184 genes over 306 genes in the category), a process in the presynapse (183/269), process in the postsynapse (131/218), synaptic signaling (106/193), metabolism (25/94) and transport (23/36) (Fig. 3e).

Beyond 57 proteins highly enriched during DA-FASS procedure, we identified a strong enrichment of the reporter protein mNeonGreen (12 unique peptides, 5.12-fold increase, adjusted $p$-value = $1.6 \times 10^{-16}$; Fig. 3f, g, Supplementary Table 3). mNeonGreen enrichment thus represents the target enrichment value for the most specific dopaminergic proteins. In line with this, the major canonical proteins involved in dopamine metabolism (Th; Ddc: DOPA decarboxylase; Slc18a2/VMAT2: Vesicular Monoamine Transporter type 2) show similar enrichment values. Slc6a3/DAT displays a slightly lower enrichment that may be

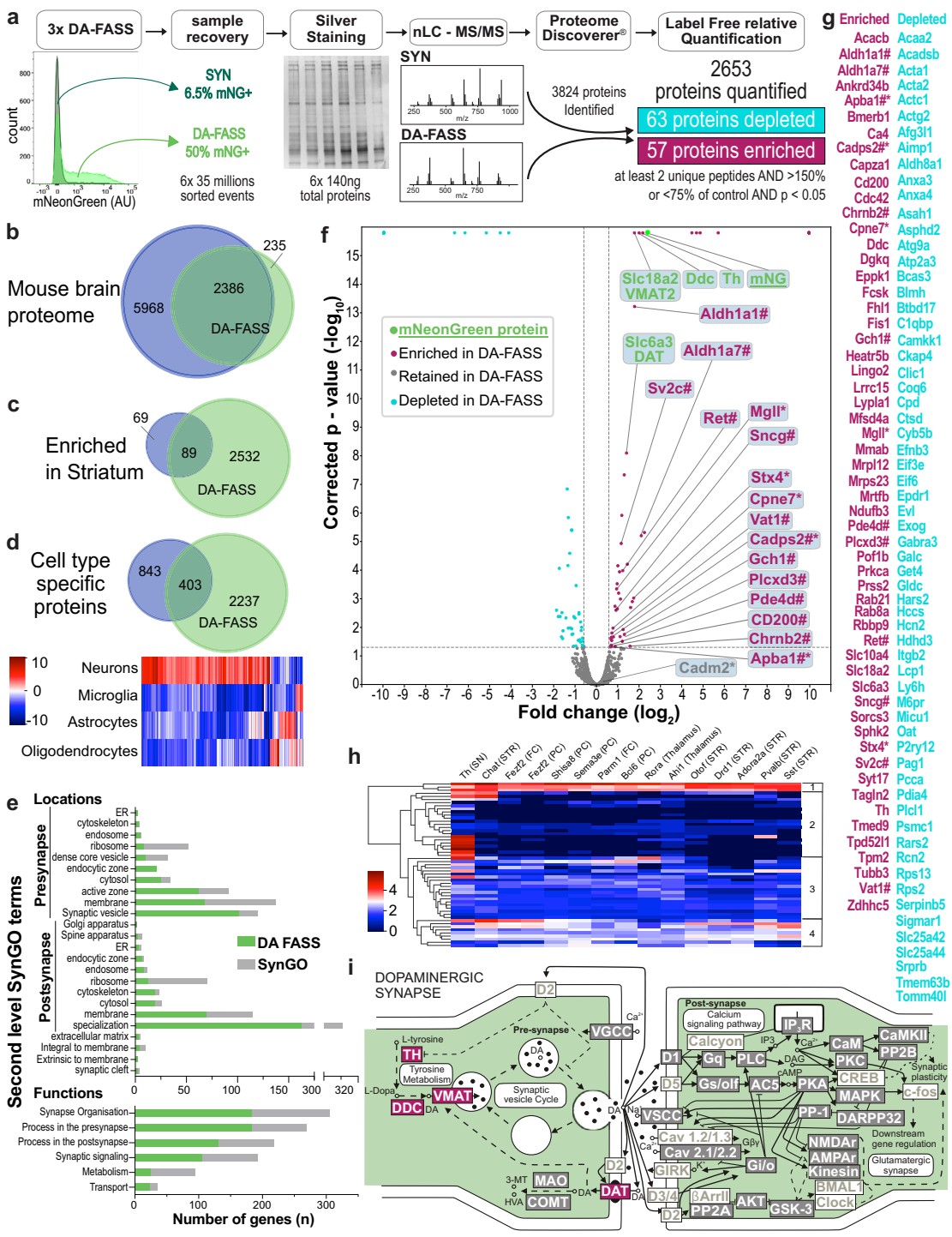

explained by the loss of DAT proteins present on the axon shaft between varicosities (Fig. 3f and Supplementary Fig. 2)[34]. Of note, 14 proteins quantified with only 2 peptides display enrichment scores much higher than the five-fold increase of our reporter. The extent of enrichment is possibly distorted by a weak detection in MS/MS. We probed the cell type expression pattern of the 57 enriched proteins with DropViz single-cell RNA sequence database. For this analysis we focused on afferent and efferent neurons in the mouse striatum[35]. This meta-analysis provides hints regarding the identity of neurons expressing the enriched markers. It defines four clusters of gene expression, from ubiquitous expression to expression restricted to Th neurons of the midbrain. This analysis further suggests that some of the DA-

FASS enriched proteins belong to various partners of the synaptic hubs (Fig. 3h and Supplementary Fig. 3). Finally, we summarized our proteome data on a model of dopaminergic transmission inspired by the KEGG database (Mmu04728)[36] to represent proteins either enriched, retained or absent from our screen (Fig. 3i Supplementary Tables 3 and 5).

**Validation of 6 new proteins enriched at dopamine synapses**. We identified a set of 12 proteins enriched in our dataset and previously shown to be important for dopamine signaling (Fig. 3f, g marked with a #). We also selected six proteins involved in membrane traffic, cell adhesion and neurotransmission, but not previously described at dopaminergic synapses, to validate

**Fig. 3 Comparative proteomic analysis of SYN and DA-FASS purified synaptosomes. a** Workflow of DA-FASS semi-quantitative proteomic analysis. Total protein content was quantified by silver staining, normalized to 140 ng for each replicate ($N = 3$) and cleaned onto SDS-PAGE gel before tryptic digestion. Proteins were analyzed by high-resolution tandem MS. Of 2653 quantified proteins, 63 were depleted in DA-FASS while 57 were enriched (Supplementary Table 3). **b–d** Venn diagrams representing the comparison of DA-FASS proteome with the mouse brain proteome[32] **b** as a whole, **c** as enriched in the striatum and **d** as cell type specific. **d** Heatmap showing cell type specific protein abundance (from[32]) among overlapping proteins. **e** Overlap analysis of the DA-FASS proteome with the second level terms of the SynGO database (mostly glutamatergic and GABAergic synapse related genes)[33]. **f** Volcano plot of DA-FASS protein quantification. Values of fold changes versus corrected p-value are plotted for each protein (on logarithmic scales). Thresholds are set at ±1.5-fold change and $p < 0.05$. Proteins are colored by subclass of canonical (green) enriched (red), depleted (cyan) and retained (grey) in the DA-FASS sample. Proteins previously described as playing a role in dopamine signaling (#). Targets selected for further experimental validations (*). Statistical significance was tested using background based two-tailed pairwise t-test adjusted using Benjamini-Hochberg correction for the false discovery rate. **g** Complete list of depleted and enriched DA-FASS proteins. **h** Heatmap of mRNA abundance of the enriched DA-FASS proteins in striatal neurons (STR) or afferent cells to the striatum (SN Substantia Nigra, Thalamus, FC Frontal Cortex, PC Posterior Cortex) (DropViz[35]). Hierarchical clustering display 4 major clusters relating to the selectivity of mRNA expression (for a more detailed heatmap see Supplementary Fig. 3). **i** Scheme of the molecular organization of a dopaminergic synapse (Adapted from ref. [36]). Enriched proteins from our DA-FASS sample are in red, retained in grey, and absent in white. Gene names for each protein class can be found in Supplementary Table 5 with absent ones greyed out.

their association with dopaminergic synapses (Fig. 3f, g marked with a *). We monitored their segregation after DA-FASS using immunofluorescence. Copine7 is a C2 domain-containing, calcium-dependent, phospholipid-binding protein (Cpne7; 1.72-fold enrichment measured in MS/MS, adjusted p-value *$p = 0.01$; see Fig. 3f and Supplementary Table 3), which displays a strong expression in dopaminergic cells of the midbrain, but also a significant expression in cholinergic interneurons (CINs) of the striatum (here labelled Interneuron Chat) and in putative cortico-striatal cells[37,38] (Fig. 4a and Supplementary Fig. 3). We find Copine7 either co-localized or apposed to Th$^+$ synaptosomes in 8% of labelled synaptosomes, a percentage that is maintained through DA-FASS (SYN: Th$^+$/Cpne7$^+$ = 8 ± 0.9%; DA-FASS: Th$^+$/Cpne7$^+$ = 8.5 ± 0.4%, $N = 3$ sorts, $n = 31$ fields of view each; SYN-DA-FASS Šídák's multiple comparison $p = 0.995$) (Fig. 4b). Mint-1/Apba1 (Mint-1 for Munc18-1 interacting protein 1, also known as Amyloid Beta Precursor Protein Binding Family A Member 1; 1.57-fold enrichment measured in MS/MS, adjusted p-value *$p = 0.044$; see Fig. 3f and Supplementary Fig. 3) is a neuronal adapter protein that interacts with the Alzheimer's disease amyloid precursor protein (APP) and plays a role at the synaptic active zone for neurotransmitter release[39,40]. Mint-1/Apba1 was also shown to be involved in amphetamine-induced dopamine release[41]. Mint-1/Apba1 mRNA displays a strong expression in Th cells of the midbrain and a milder expression in CIN and potential cortical and thalamic afferent neurons (Fig. 4a and Supplementary Fig. 3). We find Mint-1/Apba1 either co-localized or apposed with Th$^+$ particles in 4% of all labelled synaptosomes, a percentage that is increased to 8% upon DA-FASS process (SYN: Th$^+$/Mint-1$^+$ = 3.6 ± 0.5%; DA-FASS: Th$^+$/Mint-1$^+$ = 8.4 ± 0.7%, $N = 3$ sorts, $n = 30$ and 33 fields of view, respectively; SYN-DA-FASS Šídák's multiple comparison $p = 0.205$) (Fig. 4c). Cadps2 (Calcium-Dependent Activator Protein For Secretion 2; 1.62-fold enrichment measured in MS/MS, adjusted p-value *$p = 0.023$; see Fig. 3f and Supplementary Table 3) has been shown to play an important role in neurotransmitter secretion and monoamine loading in vesicles[42,43]. mRNA expression of Cadps2 is high in Th$^+$ cells and significant in putative cortico-striatal cells (Fig. 4a and Supplementary Fig. 3)[44]. Indeed, we find Cadps2 being both co-localized or apposed with Th signals in 13% of all labeled synaptosomes, a rate increased to 21% after sorting (SYN: Th$^+$/Cadps2$^+$ = 13 ± 1.7%; DA-FASS: Th$^+$/Cadps2$^+$ = 20.5 ± 1.5%, $N = 3$ sorts, $n = 30$ and 34 fields of view respectively; SYN-DA-FASS Šídák's multiple comparison *$p = 0.043$) (Fig. 4d). SynCAM 2/Cadm2 (Synaptic cell adhesion molecule 2 also known as Cell adhesion molecule 2; 1.28-fold not significant enrichment ratio) is thought to mediate heterophilic trans-synaptic adhesion at excitatory

synapses[45,46]. While SynCAM 2 mRNA is highly expressed in all populations of neurons constituting the striatal neuropil, it is striking that SynCAM 2 expression is the highest in the brain in a subcluster of Th$^+$ cells of the midbrain (Fig. 4a, Supplementary Fig. 3)[35]. Hence, SynCAM 2 represents an interesting candidate to promote synaptic adhesion at dopamine hub synapses. SynCAM 2 is mostly co-localized but also closely apposed with DAT signals in 29% of all labeled synaptosomes, a rate strongly increased to 72% after sorting (SYN: DAT$^+$/SynCAM 2$^+$ = 28.7 ± 1.6%; DA-FASS: DAT$^+$/SynCAM 2$^+$ = 71.8 ± 5.7%, $N = 3$ and $N = 2$ sorts, $n = 12$ and 10 fields of view respectively; SYN-DA-FASS Šídák's multiple comparison ****$p < 0.0001$) (Fig. 4e). Interestingly, SynCAM 2 is associated with dopamine synaptosomes at a level comparable to Th (see Fig. 2) but it is not a selective marker as it is expressed at many other synapses. Stx4 (Syntaxin-4; 3.36-fold enrichment measured in MS/MS, adjusted p-value ***$p = 0.0009$; see Fig. 3f and Supplementary Table 3) is a SNARE protein (soluble N-ethylmaleimide-sensitive factor attachment protein receptor) shown to mediate exocytosis at dendritic spines[47,48]. mRNA expression of Stx4 is moderate throughout afferent and efferent cells of the striatal neuropil (Figs. 4a and S3). Stx4 signals are mostly apposed to Th signals in 7% of all labeled synaptosomes, a rate increased to 33% after sorting (SYN: Th$^+$/Stx4$^+$ = 7.2 ± 1.2%; DA-FASS: Th$^+$/Stx4$^+$ = 33.3 ± 3.1%, $N = 3$ sorts, $n = 31$ and 30 fields of view respectively; SYN-DA-FASS Šídák's multiple comparison ****$p < 0.0001$) (Fig. 4f). Finally, Mgll (Monoglyceride lipase; 1.93-fold enrichment measured in MS/MS, adjusted p-value ***$p = 0.0004$; see Fig. 3f and Supplementary Table 3) catalyzes the conversion of monoacylglycerides to free fatty acids (+ glycerol) and is involved in the catabolism of the endocannabinoid 2-AG (2-arachidonoylglycerol)[49]. Mgll mRNA is detected at mild to high levels in most cell types afferent or efferent to the striatum, but the lowest expressers are the dopaminergic cells of the midbrain (Fig. 4a and Supplementary Fig. 3). Indeed, we find Mgll apposed to Th signals in 3% of all labelled synaptosome a percentage that increases to 10% upon sorting (SYN: Th$^+$/Mgll$^+$ = 3.22 ± 0.5%; DA-FASS: Th$^+$/Mgll$^+$ = 10.3 ± 1%, $N = 3$ sorts, $n = 33$ fields of view; SYN-DA-FASS Šídák's multiple comparison **$p = 0.002$) (Fig. 4g).

Altogether, we validated six new proteins from our screen for their selective association with dopaminergic synaptosomes. Interestingly, a comparison between MS/MS label-free quantification and immunodetection enrichment ratios reveals a linear correlation between the results (Fig. 4h).

**Proteins retained during DA-FASS delineate the association of dopaminergic varicosities in dopamine hub synapses.** To further characterize the partners in dopaminergic synaptic hubs, we

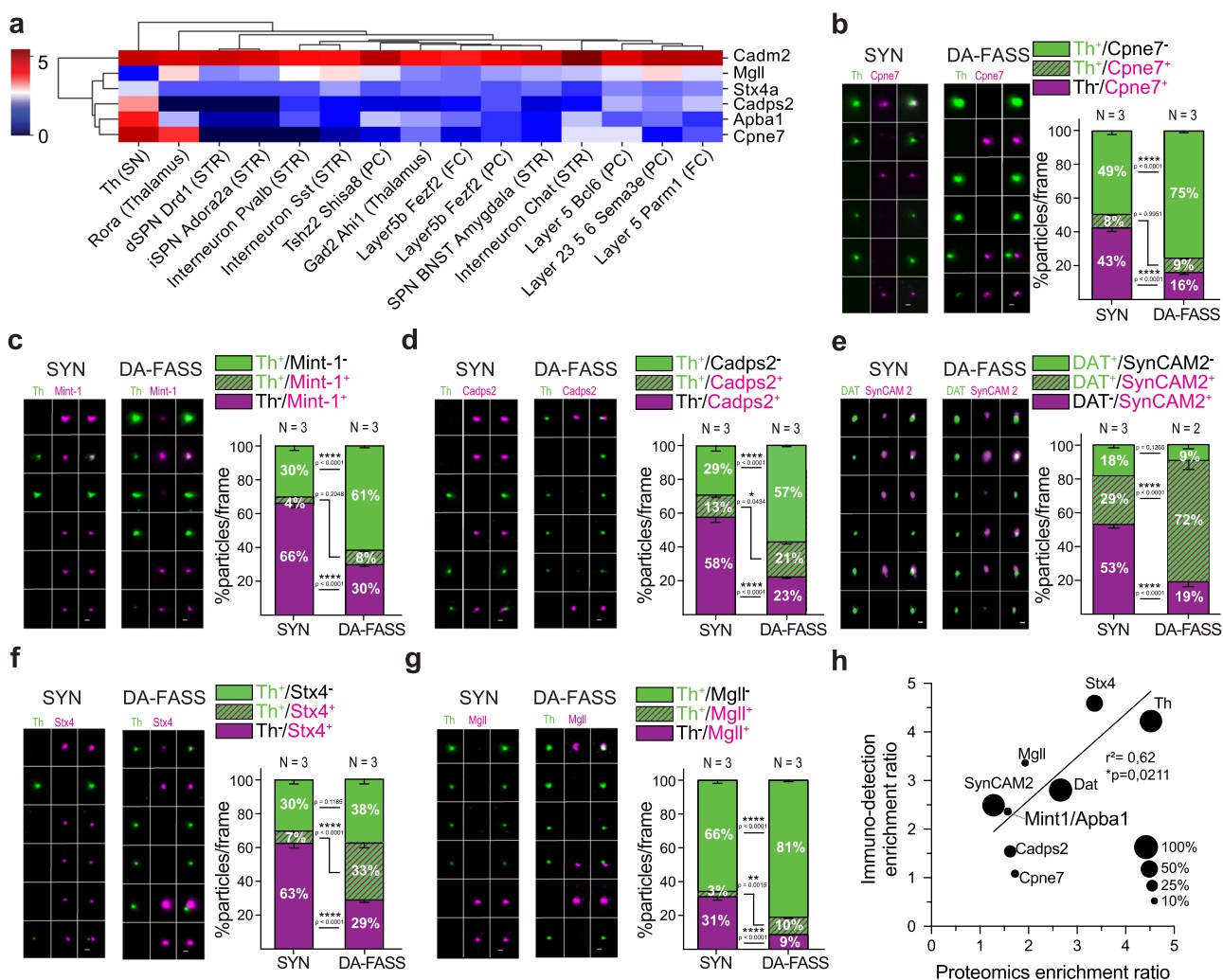

**Fig. 4 Validation of a selected set of DA-FASS enriched proteins with immunofluorescence. a** Heatmap showing cell type specific mRNA abundance of the 6 DA-FASS proteins selected for further experimental validation (detailed from Fig. 3h). **b–g** Epifluorescence images of a representative sample of synaptosome populations labelled with **b–d, f, g** anti-Th (green) or **e** DAT and **b** anti-Cpne7, **c** Mint-1/Apba1, **d** Cadps2, **e** SynCAM 2/Cadm2, **f** Stx4, **g** MgII (magenta) and analysis of staining showing particle proportions per frame. All data are mean ± SEM and pulled from $N = 2$ to $N = 3$ independent sorts and $n = 5$ to $n = 11$ field of view per independent sort. Each independent sort pooled at least 3 animals. Statistical significance was analyzed using Two-way ANOVA, **b** Th/Cpne7: Interaction $F_{2,180} = 131.9$ ****$p < 0.0001$, Condition $F_{1,180} = 0.0004$ $p = 0.984$, Immunolabelling $F_{2,180} = 570.4$ ****$p < 0.0001$; **c** Th/Mint-1: Interaction $F_{2,183} = 163.7$ ****$p < 0.0001$, Condition $F_{1,183} = 0.0009$ $p = 0.975$, Immunolabelling $F_{2,183} = 316.5$ ****$p < 0.0001$; **d** Th/Cadps2: Interaction $F_{2,187} = 110.5$ ****$p < 0.0001$, Condition $F_{1,187} = 0.004$ $p = 0.951$, Immunolabelling $F_{2,187} = 88.12$ ****$p < 0.0001$; **e** DAT/ SynCAM 2: Interaction $F_{2,60} = 84.92$ ****$p < 0.0001$, Condition $F_{1,60} = 4.371e-005$ $p = 0.995$, Immunolabelling $F_{2,60} = 73.75$ ****$p < 0.0001$ **f** Th/Stx4: Interaction $F_{2,177} = 69.25$ ****$p < 0.0001$, Condition $F_{1,177} = 0.004$ $p = 0.95$, Immunolabelling $F_{2,177} = 49.15$ ****$p < 0.0001$; **g** Th/MgII: Interaction $F_{2,192} = 98.57$ ****$p < 0.0001$, Condition $F_{1,192} = 4.217e-005$ $p = 0.995$, Immunolabelling $F_{2,192} = 1242$ ****$p < 0.0001$ with Šídák's multiple comparisons test. For all panels, scale bar = 1 µm. **h** Correlation between protein immunodetection and label-free mass spectrometry-based enrichment ratios (Two-tailed Pearson's correlation coefficient *$p = 0.021$, $r^2 = 0.62$). Correlated data are pulled from independent experiments. Dot sizes are scaled to the proportion of dopaminergic synaptosomes expressing each marker.

compared our screen with neurotransmission pathways reported in KEGG (Fig. 3i). The pathway of SV and neurotransmitter cycling shares a very high coverage with our proteome (50 protein families present out of 71 listed in the pathway; Fig. 5a grey boxed text, Supplementary Tables 3 and 5). To complete this observation, we probed for the phospho-proteins Synapsin 1&2 that are found at all presynapses[50] (abundance ratio 1.03 for both isoforms in our screen). EGFP+/Synapsin+ synaptosomes representation rises from 7 to 45% upon DA-FASS (SYN: EGFP+/Synapsin+ = 6.6 ± 1.2%; DA-FASS: EGFP+/Synapsin+ = 44.9 ± 3.3%, $N = 2$ and $N = 3$ sorts, $n = 9$ and 10 fields of view respectively; SYN-DA-FASS Šídák's multiple comparison ****$p < 0.0001$; Fig. 5b) while EGFP −/Synapsin+ synaptosomes are reduced from 83 to 34% after sort

(SYN: EGFP−/Synapsin+ = 83.2 ± 1.1%; DA-FASS: EGFP−/Synapsin+ = 33.8 ± 2.4%, $N = 2$ and $N = 3$ sorts, $n = 9$ and 10 field of view, respectively; SYN-DA-FASS Šídák's multiple comparison ****$p < 0.0001$). (Fig. 5b).

We then explored the proteome related to excitatory synapses. Our coverage is reliable because most categories of proteins are kept after DA-FASS (78 out of 128 listed proteins Fig. 5c grey boxed text, Supplementary Tables 3 and 5). We immunolabelled DA-FASS synaptosomes for the 2 vesicular glutamate transporters (VGLUT). VGLUT1 is expressed by excitatory cortico-striatal inputs while thalamo-striatal inputs express VGLUT2; both input impinge on spines of the spiny projection neurons (SPNs)[5,51]. VGLUT1 varicosities are opposed to EGFP

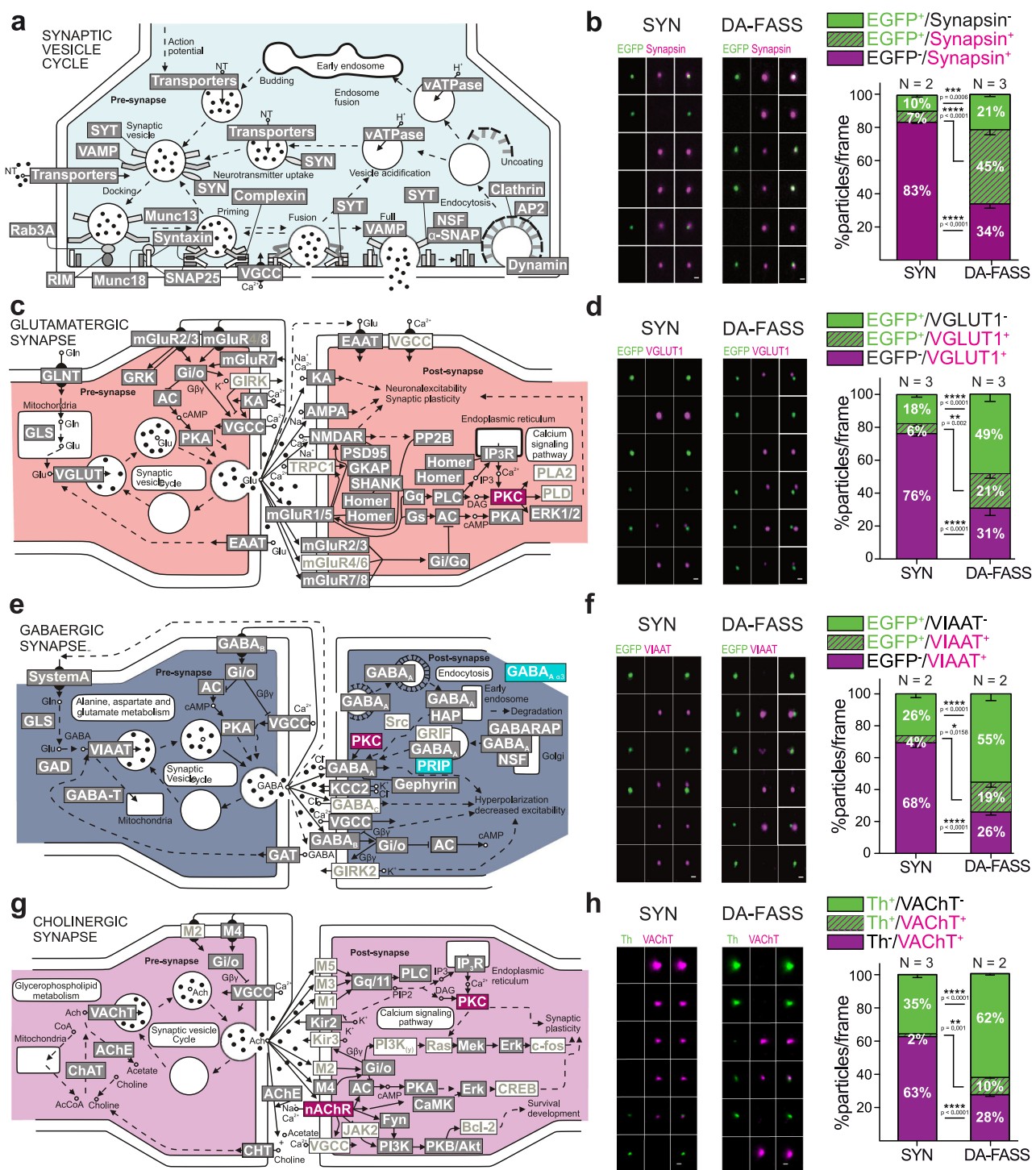

**Fig. 5 Proteomics and immunofluorescence of DA-FASS sample reveals dopamine synapse association with other synaptic partners. a** Scheme of the molecular organization of the synaptic vesicle cycle, **c** glutamatergic, **e** GABAergic and **g** cholinergic synapses (Adapted from the database KEGG). Proteins enriched in DA-FASS samples are in red, depleted in cyan, retained in grey, and absent in white. Gene names for each protein can be found in Supplementary table 4 (ST4). **b, d, f, h** Epifluorescence images of a representative sample of synaptosome populations before and after sorting labelled with anti-EGFP or anti-Th (green) and **b** anti-Synapsin, **d** VGLUT1, **f** VIAAT and **h** VAChT (magenta). Quantification of stainings showing particle proportions per frame. All data are mean ± SEM and pulled from $N = 2$ to $N = 3$ independent sorts and $n = 4$ to $n = 11$ field of view per independent sort. Each independent sort pooled at least 3 animals. **b** EGFP/Synapsin: Interaction $F_{2,51} = 237,8$ ****$p < 0.0001$, Condition $F_{1,51} = 0.01$ $p = 0.92$, Immunolabelling $F_{2,51} = 237.5$ ****$p < 0.0001$. **d** EGFP/VGLUT1 Interaction $F_{2,63} = 91.49$ ****$p < 0.0001$, Condition $F_{1,63} = 0.002$ $p = 0.97$, Immunolabelling $F_{2,63} = 92.06$ ****$p < 0.0001$. **f** EGFP/VIAAT: Interaction $F_{2,78} = 54.90$ ****$p < 0.0001$, Condition $F_{1,78} = 0.04$ $p = 0.844$, Immunolabelling $F_{2,78} = 55.34$ ****$p < 0.0001$. **h** Th/VAChT Interaction $F_{2,144} = 180.3$ ****$p < 0.0001$, Condition $F_{1,144} = 0.016$ $p = 0.9$, Immunolabelling $F_{2,144} = 412.2$ ****$p < 0.0001$ with Šídák's multiple comparisons test. For all panels, scale bar $= 1\,\mu m$. See VGLUT2 immunofluorescence analysis in Supplementary Fig. 4.

varicosities at hub synaptosomes (see ROI gallery; Fig. 5d). Through DA-FASS, EGFP$^-$/VGLUT1$^+$ synaptosomes are depleted more than two-fold (SYN: EGFP$^-$/VGLUT1$^+$ = 76.2 ± 1.6%; DA-FASS: EGFP$^-$/VGLUT1$^+$ = 30.9 ± 4.6%, $N$ = 3 sorts, $n$ = 12 and 11 fields of view respectively; SYN-DA-FASS Šídák's multiple comparison ****$p$ < 0.0001; Fig. 5d). Yet, 30% of dopaminergic EGFP$^+$ synaptosomes (21/(21 + 49)) are associated with a VGLUT1 presynapse (DA-FASS: EGFP$^+$/VGLUT1$^+$ = 20.8 ± 2.7%, $N$ = 3 sorts, $n$ = 11 fields of view; Fig. 5d) and enriched three-fold through DA-FASS (from 6 to 21%; SYN: EGFP$^+$/VGLUT1$^+$ = 5.9 ± 1.4%, $N$ = 3 sorts, $n$ = 12 fields of view; SYN-DA-FASS Šídák's multiple comparison **$p$ = 0.002; Fig. 5d). VGLUT2 signals follow the same trend even though VGLUT2 is less associated with EGFP$^+$ synaptosomes than VGLUT1 (SYN: EGFP$^+$/VGLUT2$^+$ = 1.8 ± 1.1%, DA-FASS: EGFP$^+$/VGLUT2$^+$ = 11.2 ± 1.4%, $N$ = 2 and $N$ = 3 sorts, $n$ = 6 and $n$ = 14 fields of view, respectively; Supplementary Fig. 4). As a negative control, we assessed whether VGLUT1 and VGLUT2 synaptosomes may co-purify through FASS by performing FASS sorting and labelling of VGLUT1$^{venus}$ striatal synaptosomes (Supplementary Fig. 5a–g). As expected, VGLUT2-synaptosomes are mostly segregated from VGLUT1$^{venus}$ synaptosomes, and VGLUT2$^+$/VGLUT1$^{Venus+}$ particles are not co-enriched through VGLUT1-FASS sorting (9% in SYN sample vs 6% in VGLUT1-FASS synaptosomes; Supplementary Fig. 5f–g upper right quadrants). The absence of association is consistent with the fact that these 2 markers were shown to contact distinct spines on SPNs[5,52,53].

Our proteome also displays an abundant representation of markers of inhibitory synapses kept through DA-FASS enrichment (69 out of 106 listed proteins, Fig. 5e gray boxed text, Supplementary Tables 3 and 5). Of note, two proteins of GABAergic synapses are depleted after DA-FASS (Gabra3 and Prip; Fig. 5e blue boxed text, Supplementary Tables 3 and 5). We therefore probed DA-FASS synaptosomes for the vesicular inhibitory amino-acid transporter (VIAAT), which labels GABAergic terminals arising from all inhibitory neurons of the striatum[54]. EGFP$^+$/VIAAT$^+$ hub synaptosomes display almost 5-fold enrichment through DA-FASS (SYN: EGFP$^+$/VIAAT$^+$ = 4.1 ± 0.9%, DA-FASS: EGFP$^+$/VIAAT$^+$ = 18.8 ± 3.2%, $N$ = 2 sorts, $n$ = 9 and $n$ = 19 fields of view respectively, SYN-DA-FASS Šídák's multiple comparison *$p$ = 0.016; Fig. 5f), while the EGFP$^-$/VIAAT$^+$ population is depleted more than 2-fold (SYN: EGFP$^-$/VIAAT$^+$ = 68.2 ± 3.2 %, DA-FASS: EGFP$^-$/VIAAT$^+$ = 25.9 ± 1.9%, $N$ = 2 sorts, $n$ = 9 and $n$ = 19 fields of view, respectively, SYN-DA-FASS Šídák's multiple comparison ****$p$ < 0.0001; Fig. 5f). Hence, GABAergic synaptosomes are associated with 26% of the dopaminergic synaptosomes (19/(19 + 55) × 100; Fig. 5f).

Finally, striatal neuropils harbour a dense cholinergic innervation by local CINs that function in tight interrelation with dopaminergic signals[55,56]. In accordance, our proteome also displays a significant fraction of cholinergic markers that are kept throughout DA-FASS (50 out 112 listed proteins; Fig. 5g grey boxed text, Supplementary Tables 3 and 5). The beta2 nicotinic receptor subunit (Chrnb2) is even significantly enriched (2.98-fold enrichment measured in MS/MS, adjusted $p$-value *$p$ = 0.048; Fig. 5g red boxed text, Supplementary Tables 3 and 5). Indeed, it was shown to mediate cholinergic signalling onto dopaminergic varicosities[57]. To confirm a physical binding of dopaminergic varicosities with cholinergic ones, we probed for the Vesicular Acetyl Choline Transporter (VAChT). VAChT signals is occasionally seen apposed to Th positive dots with a 6-fold increase through DA-FASS enrichment (SYN: Th$^+$/VAChT$^+$ = 1.7 ± 0.4%, DA-FASS: Th$^+$/VAChT$^+$ = 10.2 ± 0.7%, $N$ = 3 and $N$ = 2 sorts, $n$ = 30 and $n$ = 20 fields of view respectively, SYN-DA-FASS Šídák's multiple comparison **$p$ = 0.001; Fig. 5h). Through DA-

FASS, Th$^-$/VAChT$^+$ synaptosomes are depleted nearly two-fold (SYN: Th$^-$/VAChT$^+$ = 62.5 ± 2.1%, DA-FASS: Th$^-$/VAChT$^+$ = 28 ± 1.3%, $N$ = 3 and $N$ = 2 sorts, $n$ = 30 and $n$ = 20 fields of view, respectively, SYN-DA-FASS Šídák's multiple comparison ****$p$ < 0.0001; Fig. 5h).

Hence, our proteomic and immunofluorescence data support a very frequent association of dopamine presynapses with all the major synaptic partners operating in striatal neuropil. This finding further validates our earlier electron microscopy observations (Fig. 1j–l). To validate the accuracy and specificity of our results, we performed several controls. A random associations test was applied to our images in order to establish the probability for separate particles to sediment at the same sites by chance (see methods). Indeed, for all our datasets, random associations occur on less than 0.5% of all events while we observe at least 10% for synaptic hub-related associations in sorted samples (see Table 1). As a final control for the specificity of hub-synaptosome adhesion, we performed an additional VGLUT1-FASS experiment in which we selectively sorted aggregates and large events to analyze them with electron microscopy (Supplementary Fig. 5h–j). Upon reanalysis, sorted aggregates display a strong increase in the representation of small and large aggregates (Supplementary Fig. 5h, i). Singlets are still strongly represented in the reanalyzed sample as it is common to break down aggregates into singlets through the shearing forces applied in the nozzle of the sorter (Supplementary Fig. 5h, i). Electron micrographs display profiles of large particles (3–6 μm in diameter) that are difficult to relate to identifiable features of the tissue and very different from the DA-FASS synaptosomes displayed in Fig. 1 (Supplementary Fig. 5j).

Altogether, we identified the association of dopaminergic with glutamatergic and GABAergic synapses in synaptic hub structures that may mediate the modulatory influence of dopamine over excitatory and inhibitory synaptic signalling. Cholinergic inputs from CINs may also take part in this association.

**Spatial organization of dopaminergic synaptic hubs.** We analyzed the relative position of each marker to dopaminergic varicosities (immunolabelled for EGFP, Th or DAT) by measuring the centre to centre distance with the other markers used above on the whole population of synaptosomes imaged by wide-field microscopy. Th is co-localized with EGFP and seen at an average distance of 0.174 μm (0.174 ± 0.003 μm, $n$ = 1175) close to the resolution of the epifluorescence setup (0.250 μm), while the most distant marker, D1R, is apposed on average at 0.513 μm (0.513 ± 0.022 μm, $n$ = 246) from the EGFP$^+$ centre (see Fig. 6a). Next to the dopaminergic varicosity (285–300 nm), we find SynCAM 2 (0.285 ± 0.006 μm, $n$ = 949) and D2R (0.2982 ± 0.015 μm, $n$ = 181). Slightly more distant (318–408 nm), we find the presynaptic cholinergic transporter VAChT (0.318 ± 0.017 μm, $n$ = 151), the Glutamatergic transporter

**Table 1 Observed versus simulated random associations of immunolabeled markers.**

| Immunolabelling | Observed associations (%) | Simulated random associations (%) |
|---|---|---|
| EGFP$^+$ VGLUT1$^+$ | 20.8 | 0.5 |
| EGFP$^+$ VGLUT2$^+$ | 11.2 | 0.5 |
| EGFP$^+$ VIAAT$^+$ | 18.8 | 0.5 |
| VGLUT1$^{Venus+}$ VGLUT2$^+$ | 5.5 | 0.4 |
| VGLUT1$^{Venus+}$ Th$^+$ | 31.5 | 0.5 |
| Th$^+$ VAChT$^+$ | 10.2 | 0.5 |

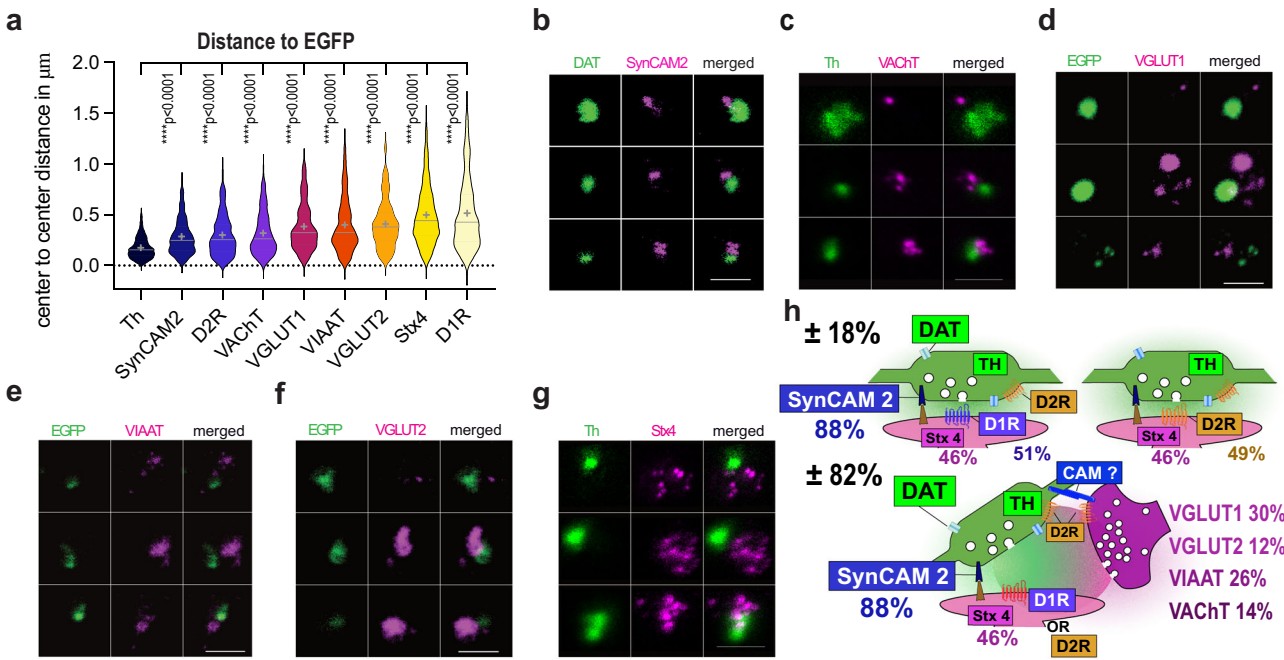

**Fig. 6 Modeling the spatial organization of dopaminergic synaptic hubs. a** Distance to dopamine varicosity center for all stained proteins in increasing order of average distance (EGFP reference for Th, D1R, D2R, VIAAT, VGLUT1 and -2; Th reference for VAChT, Stx4; DAT reference for SynCAM 2). Distance between EGFP and Th or DAT is not resolved by conventional epifluorescence microscopy (below 250 nm). Violin plot data are displayed with 25–75th percentiles, median as a center line, and mean as a cross. Mean ± SEM are from $N = 2$ to $N = 3$ independent sorts pooling 3 animals each with n the number of distances analyzed in double positive ROIs (Th = 0.174 ± 0.003, $n = 1175$; SynCAM 2 = 0.285 ± 0.006, $n = 949$; D2R = 0.298 ± 0.015, $n = 181$; VAChT = 0.318 ± 0.017, $n = 151$; VGLUT1 = 0.381 ± 0.017, $n = 193$; VIAAT = 0.398 ± 0.015, $n = 362$; VGLUT2 = 0.408 ± 0.019, $n = 150$; Stx4 = 0.497 ± 0.009, $n = 880$; D1R = 0.513 ± 0.022, $n = 246$. Kruskal–Wallis test, ****$p < 0.0001$ with Dunn's multiple comparisons was performed). **b–g** STED images of synaptosomes stained for **b** DAT, **c–g** Th or **d–f** EGFP (green) and, **b** SynCAM 2, **c** VAChT, **d** VGLUT1, **e** VIAAT, **f** VGLUT2, **g** Stx4, (magenta) coming from one experiment. Scale bars 1 μm. **h** Synaptomic model of the dopamine synapse population in the striatum. Based on the 51% D1R-positive and 78% D2R-positive EGFP + synaptosomes identified in Fig. 2 we infer that dopaminergic synaptosomes comprise varicosities apposed to a postsynaptic element with either D1R (51%) or D2R (49%). D2R also being represented at the presynapse. Stx4 is present at 46% of postsynapses (see Fig. 4). Present in nearly all EGFP + boutons (88 %) SynCAM 2 represents a good candidate for synaptic adhesion. Up to 82% of dopamine varicosities form a synaptic hub with excitatory, inhibitory or cholinergic synapses (VGLUT1 + in 30%, VGLUT2 + in 12%, VIAAT+ in 26%; VAChT+ in 14%; see Fig. 5).

VGLUT1 (0.381 ± 0.017 μm, $n = 193$), the GABAergic transporter VIAAT (0.398 ± 0.015 μm, $n = 362$) and the Glutamatergic transporter VGLUT2 (0.408 ± 0.019 μm, $n = 150$). Finally, Stx4 (0.497 ± 0.009 μm, $n = 880$) the presumed spine associated snare protein and the postsynaptic D1R display the most distant apposition to Th⁺ and EGFP⁺ varicosities, respectively (Fig. 6a). We confirmed the order of distances with high spatial resolution STED microscopy for a selection of markers (SynCAM 2, VAChT, VGLUT1, VIAAT, VGLUT2, Stx4; Fig. 6b–g).

From the data gathered in Figs. 2, 4–6 we propose a model of the dopaminergic projection in which most dopaminergic varicosities adhere to postsynaptic elements labelled by either D1R or D2R with the presence of SynCAM 2 and Stx4. In addition, around 80% of dopaminergic synapses are also associated with other presynapses in synaptic hub structures clearly identified in electron and STED microscopy. Dopamine hub synapses are formed with cholinergic (14%), GABAergic (26%) and Glutamatergic synapses (42% VGLUT1 cortico-striatal + VGLUT2 Thalamo-striatal; Fig. 6h).

**Comparison of VGLUT1 excitatory cortico-striatal hub versus regular synapse.** Finally, we questioned whether the association with a dopaminergic input correlates with changes at glutamatergic synapses. To that end, we sorted striatal VGLUT1venus synaptosomes using VGLUT1-FASS (Fig. 7a and Supplementary Fig. 5a–e), stained them with Th to classify them into Th⁺ and

Th⁻, and probed several markers of glutamatergic synapses. In this experiment, we confirm the presence of VGLUT1⁺/Th⁺ hub synapses: they are enriched with FASS and represent 41% (32/(32 + 45)) of the VGLUT1⁺ population (SYN: VGLUT1Venus +/Th⁺ = 15.5 ± 2.6%, $N = 2$ sorts and $n = 19$ fields of view, VGLUT1-FASS: VGLUT1Venus+/Th⁺ = 31.5 ± 4%, VGLUT1venus+/Th⁻ = 45.5 ± 3.2%, $N = 2$ sorts, $n = 20$ fields of view, SYN-VGLUT1-FASS Šídák's multiple comparison ***$p = 0.0006$; Fig. 7b). We further confirmed the association between VGLUT1venus and Th using multiplexed capillary electrophoresis-based immunoblots with both detections in the same capillary. Th levels are maintained after FASS (SYN: Th = 4.37.10⁵ ± 0.79.10⁵, VGLUT1-FASS, Th = 3.58.10⁵ ± 1.08.10⁵, $N = 3$ sorts, unpaired t-test $p = 0.584$) while VGLUT1venus is enriched (SYN: VGLUT1venus = 3.92.10⁵ ± 0.60.10⁵, VGLUT1-FASS: VGLUT1venus = 6.70.10⁵ ± 0.74.10⁵, $N = 3$ sorts; unpaired t-test *$p = 0.043$; Fig. 7c). VGLUT1venus labels the cluster of SVs and is a proxy to the loading of SV with glutamate[58]. We find a significantly higher VGLUT1venus signal in Th⁺/VGLUT1⁺ compared to Th⁻/VGLUT1⁺ synaptosomes (Th⁻: VGLUT1venus = 1.43.10⁵ ± 0.02.10⁵ $N = 3$ $n = 3609$; Th⁺: VGLUT1venus = 1.83.10⁵ ± 0.05.10⁵, $N = 3$ sorts n = 1206 particles; Th⁻-Th⁺ ****$p < 0.0001$ Mann–Whitney test; see Fig. 7d). Bassoon is a scaffold protein of the active zone of neurotransmitter release present at most glutamatergic and GABAergic synapses[59]. In a recent report Bassoon was described to be present at only a third

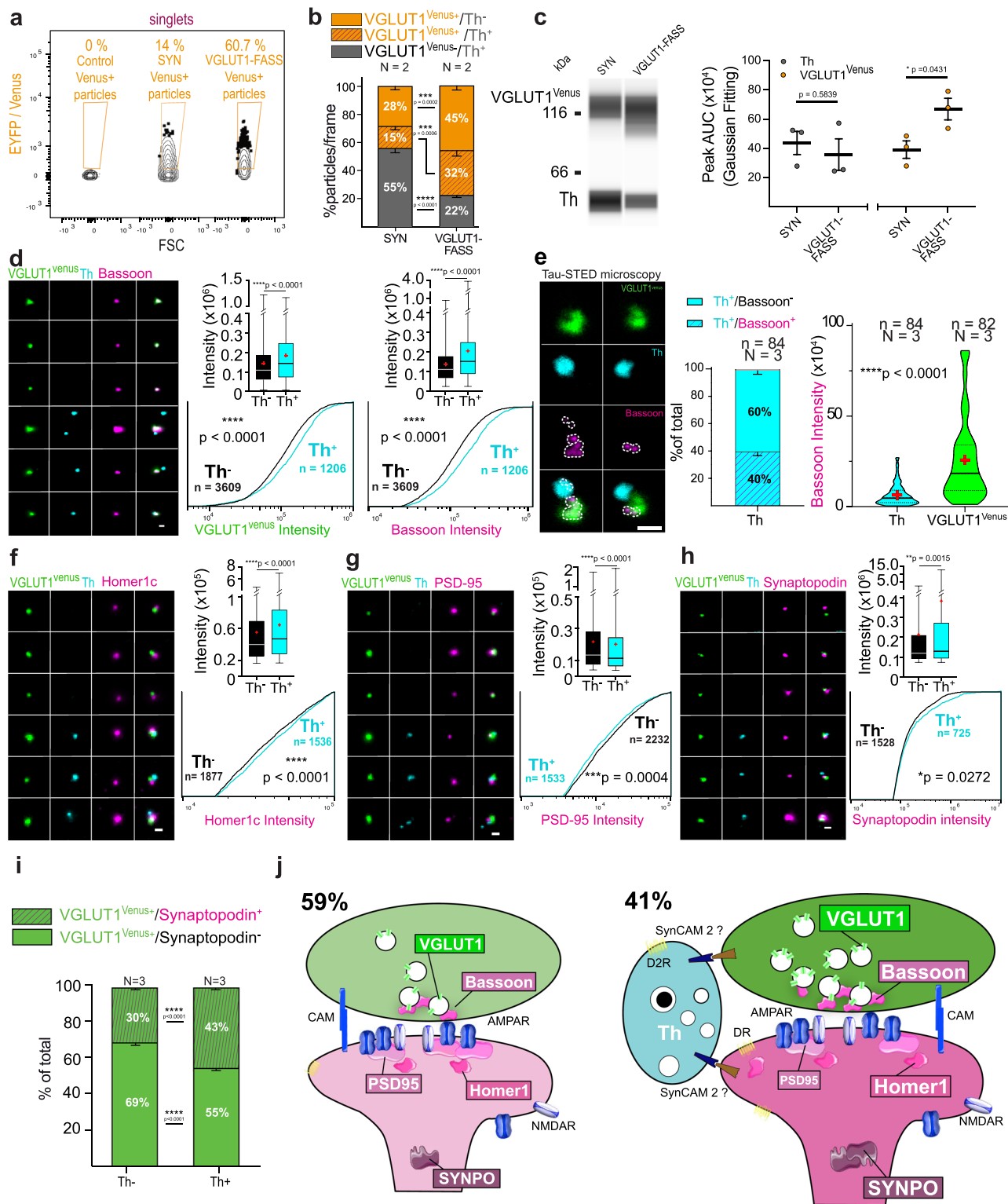

of dopamine varicosities[13]. Here, we confirm that most VGLUT1-venus+ synaptosomes display Bassoon signal (VGLUT1Venus+/Bassoon+ = 93.4 ± 0.4%, $n = 4456$, $N = 4$ sorts; Supplementary Fig. 6a), however, only 20% of Th+/ VGLUT1venus− elements contained a Bassoon cluster (VGLUT1−/Th+ = 19.6 ± 3.7, $n = 45$, $N = 3$ sorts; Supplementary Fig. 6a). By measuring Bassoon signal intensity at VGLUT1 synapses in epifluorescence images, we find a 1.5-fold higher bassoon signal in Th+/ VGLUT1venus+ compared to Th−/ VGLUT1venus+ synaptosomes (Th−/VGLUT1venus+:

Bassoon = 1.38.10^5 ± 0.02.10^5 $N = 3$ sorts and $n = 3609$ particles; Th+/VGLUT1venus+: Bassoon = 2.06.10^5 ± 0.07.10^5 $N = 3$ sorts and $n = 1206$ particles; Th−/VGLUT1venus+-Th+/VGLUT1venus+ ****$p < 0.0001$ Mann–Whitney test; Fig. 7d and Supplementary Fig. 6a). As Bassoon increase may be due to its presence within Th varicosities, we quantified STED images to discriminate the origin of Bassoon signal in dopamine hub synapses. In hubs, we observe 60% of Th+ varicosities devoid of a Bassoon cluster (Hub_Th_V-GLUT1: Th+/Bassoon+ = 39.7 ± 4.1%, $n = 84$ synaptosomes,

**Fig. 7 Molecular remodeling at cortico-striatal dopamine hub synapse. a** striatal VGLUT1-FASS. SYN show 14% of VGLUT1$^{Venus+}$ synaptosome enriched to 60.7% in VGLUT1-FASS. **b** Immunofluorescence population analysis of SYN and VGLUT1-FASS (mean ± SEM from $N = 2$ sorts (3 animals each) with $n = 9$ to $n = 10$ fields of view per sort; Two-way ANOVA, VGLUT1$^{Venus}$/Th: Interaction $F_{2,111} = 47.14$ ****$p < 0.0001$, Condition $F_{1,111} = 0.003$ $p = 0.96$, Immunolabelling $F_{2,111} = 15.54$ ****$p < 0.0001$). Note that 41% of VGLUT1$^{venus+}$ synaptosomes are dopamine hub synapses ($41 = 32 / (32 + 45)$; examples in **d-h**). **c** Immunoblot against VGLUT1$^{Venus}$ and Th. Averaged Th (grey) and VGLUT1$^{Venus}$ (orange) peak AUC (mean ± SEM from $n = 3$; Two-way ANOVA, Th/ VGLUT1$^{Venus}$: Interaction $F_{1,8} = 4.757$ $p = 0.061$, Protein $F_{1,8} = 1.468$ $p = 0.26$, Condition $F_{1,8} = 2.656$ $p = 0.142$); for uncropped immunoblots see supplementary material or original data files in source data. **d, f, g, h** VGLUT1-FASS immunolabelled for VGLUT1$^{Venus}$ (green), Th (cyan), and **d** Bassoon, **f** Homer1c, **g** PSD-95 and **h** synaptopodin (magenta). Comparison of Th$^+$/VGLUT1$^{venus+}$ hub synapse versus Th$^-$/VGLUT1$^{venus+}$ synapse staining intensity (data represented as min/max, mean (red cross) and median (center line), $N = 3$; Two-tailed Mann–Whitney test) **d** VGLUT1$^{Venus}$: ****$p < 0.0001$; Bassoon: ****$p < 0.0001$; **f** Homer1c: ****$p < 0.0001$; **g** PSD-95: ****$p < 0.0001$; **h** Synaptopodin: **$p = 0.002$; and Kolmogorov–Smirnov test for CDF **d** VGLUT1$^{Venus}$: ****$p < 0.0001$; Bassoon: ****$p < 0.0001$; **f** Homer1c: ****$p < 0.0001$; **g** PSD-95: ***$p = 0.0004$; **h** Synaptopodin: *$p = 0.027$. **e** STED images of VGLUT1$^{Venus}$/Th/Bassoon synaptosomes. Detection and quantification of Bassoon in Th varicosities of hubs (Mean ± SEM $N = 3$; Mann–Whitney test, Bassoon: ****$p < 0.0001$). **i** Detection of Synaptopodin$^+$ staining at dopamine hub synapses (Mean ± SEM $N = 3$; Two-way ANOVA, VGLUT1$^{Venus}$/Synaptopodin: Interaction $F_{1'320} = 95.48$ ****$p < 0.0001$, Condition $F_{1,320} = 0.169$ $p = 0.682$, Immunolabelling $F_{1,320} = 326$ ****$p < 0.0001$). **j** Synaptomic model of VGLUT1 striatal synapses. 41% are VGLUT1 dopamine hub synapses. Dopamine hub synapses are remodeled as shown by an increased intensity of VGLUT1, Bassoon, Homer1, Synaptopodin and a decrease of PSD-95 compared to regular VGLUT1 synapses. Scale bar = 1 μm.

Th$^+$/Bassoon$^-$ = 60.3 ± 4.055, $n = 82$ synaptosomes; $N = 3$ sorts; Fig. 7e). In the minority of Th$^+$/Bassoon$^+$ synapses, the Bassoon signal intensity represents only a quarter of that measured for Bassoon within VGLUT1$^{Venus+}$ synaptosomes (Hub_Th_VGLUT1; In Th: Bassoon = 0.65.10$^5$ ± 0.07.10$^5$, $n = 84$ synaptosomes, In VGLUT1$^{Venus}$: Bassoon = 2.55.10$^5$ ± 0.25.10$^5$, $n = 82$ synaptosomes, $N = 3$ sorts; Th-VGLUT1$^{Venus}$ ****$p < 0.0001$ Mann–Whitney; Fig. 7e). Hence, we conclude that most of the increase of Bassoon signal at dopamine hub synapses is occurring within the VGLUT1 terminals under the influence of dopaminergic innervation. RIM1 is another active zone scaffold protein which is essential for dopamine release[13]. We monitored the presence of RIM1 at Th$^+$ varicosities of synaptic hubs. Indeed, we confirm the presence of RIM1 in 70% of Th$^+$ varicosities (Th$^+$/RIM1$^+$ 69.7 ± 4.3, Th$^+$/RIM1$^-$ 30.3 ± 4.3 $n = 73$, $N = 3$ sorts; Supplementary Fig. 6c) with a lower intensity than in neighbouring VGLUT1 boutons (Hub_Th_VGLUT1; In Th: RIM1 = 0.32 ± 0.03.10$^5$, $n = 73$, In VGLUT1: RIM1 = 2.55.10$^5$ ± 0.25.10$^5$, $n = 74$, $N = 3$ independent sorts; RIM1 in Th-RIM1 in VGLUT1 Mann–Whitney test ***$p = 0.0009$; Supplementary Fig. 6c).

We then focused on postsynaptic proteins. Homer1c is a calcium-binding scaffold protein important for metabotropic glutamate receptor signalling[60]. We find higher Homer1c signal in Th$^+$/VGLUT1$^{Venus+}$ compared to Th$^-$/VGLUT1$^{Venus+}$ synaptosomes (Th$^+$/VGLUT1$^{Venus+}$: Homer1c = 0.55.10$^5$ ± 0.01.10$^5$ $n = 1877$ particles; Th$^-$/VGLUT1$^{Venus+}$: Homer1c = 0.65.10$^5$ ± 0.01.10$^5$, $n = 1536$ particles, $N = 3$ independent sorts; Th$^+$/VGLUT1$^{Venus+}$-Th$^-$/VGLUT1$^{Venus+}$ ****$p < 0.0001$ Mann–Whitney; Fig. 7f). In contrast, we find that the signal for PSD-95, a major postsynaptic density scaffold[61], is slightly decreased in Th$^+$/VGLUT1$^{Venus+}$ compared to Th$^-$/VGLUT1-$^{Venus+}$ synaptosomes (Th$^-$/VGLUT1$^{Venus+}$: PSD-95 = 0.22.10$^5$ ± 0.005.10$^5$, $n = 2232$ particles; Th$^+$/VGLUT1$^{Venus+}$: PSD-95 = 0.20.10$^5$ ± 0.006.10$^5$, $n = 1533$ particles, $N = 3$ sorts; Th$^-$/VGLUT1$^{Venus+}$-Th$^+$/VGLUT1$^{Venus+}$ ****$p < 0.0001$ Mann–Whitney; Fig. 7g). To further characterize the postsynaptic compartment, we labelled Synaptopodin (Synpo) a marker of the spine apparatus[62]. The spine apparatus and Synpo are found at a minority of spines in the forebrain and is thought to be involved in structural plasticity[63]. Synpo is increased almost two-fold at Th$^+$/VGLUT1$^{Venus+}$ compared to Th$^-$/VGLUT1$^{Venus+}$ synaptosomes (Th$^-$/VGLUT1$^{Venus+}$: Synpo = 2.14.10$^5$ ± 0.06.10$^5$, $n = 1528$ particles; Th$^+$/VGLUT1$^{Venus+}$: Synpo = 3.83.10$^5$ ± 0.37.10$^5$ $n = 725$ particles, $N = 3$ sorts; Th$^-$/VGLUT1$^{Venus+}$-Th$^+$/VGLUT1$^{Venus+}$ **$p = 0.002$ Mann–Whitney; Fig. 7h). Surprisingly, we also observe that Synpo is relatively well maintained

through FASS purification in the striatum while we had previously shown a strong depletion in forebrain samples[27,64]. In fact, Synpo is among the markers seen specifically enriched in the striatum in mass spectrometry compared to other brain regions[32]. In our LC-MS/MS screen, Synpo is unchanged after FASS (abundance ratio of 1 with 17 unique peptides, adjusted $p$-value $p = 1$; Supplementary Table 3), a trend we confirm with immunofluorescence probing Synpo on DA-FASS samples (Supplementary Fig. 6b). In the triple staining experiment with VGLUT1-FASS, Synpo signals are apposed to VGLUT1 positive dots more frequently when Th$^+$ varicosities are present (VGLUT1-FASS: VGLUT1$^{Venus+}$/Synpo$^+$/Th$^+$ = 43.2 ± 1.6%, $n = 81$ particles; VGLUT1$^{Venus+}$/Synpo$^+$/Th$^-$ = 30.2 ± 0.9%, $n = 81$ particles, $N = 3$ sorts; VGLUT1$^{Venus+}$/Synaptopodin$^+$/Th$^+$-VGLUT1$^{Venus+}$/Synaptopodin$^+$/Th$^-$ Šídák's multiple comparison ****$p < 0.0001$; Fig. 7i).

Finally, we propose a model that summarizes our findings regarding dopamine hub synapses involving VGLUT1 (Fig. 7j). Altogether, our results show that a selective set of markers of SV cluster, active zone, postsynaptic density and spine apparatus at VGLUT1 synapses on SPNs display a significant increase upon innervation by dopaminergic varicosities. This observation strengthens the notion that dopamine hub synapses represent a mechanically resilient functional structure.

## Discussion

To unravel specific molecular and cellular features of modulatory neurotransmission, we targeted the dopaminergic projection from the substantia nigra and ventral tegmental area to the striatum using FASS[18,27]. Specificity for dopaminergic synaptosomes was validated by the enrichment for presynaptic dopaminergic markers as well as the adhesion of dopaminergic varicosities to postsynaptic elements containing either D1R or D2R. We produced a proteome with proteins significantly enriched through DA-FASS purification and validated the enrichment and localization of 6 of them. We show the association of dopaminergic synapses with glutamatergic, GABAergic and cholinergic synapses in "dopamine hub synapses" that connect tightly several synapses together. Finally, we observed that innervation of glutamatergic synapses by dopaminergic varicosities correlates with a molecular strengthening of the whole synapse.

**A proteome of dopaminergic synapses in the striatum.** The molecular characterization of FASS dopaminergic synaptosomes

quantified 2653 proteins between unsorted synaptosomes and DA-FASS samples. Sixty three proteins are significantly depleted through DA-FASS while 57 proteins are found strongly enriched. Hence, most proteins are kept in the process. This may be attributed on the one hand to the relative low purity of our DA-FASS samples until now (more than 40% of EGFP- singlets are left after sorting according to reanalysis, Figs. 1 and 3), and on the other hand, to the existence of dopamine hub synapses with most other neuronal partners involved in striatal networks. Yet, the enrichment factor for dopaminergic synaptosomes is quite high (between 5- and 10-fold), allowing the detection of proteins selectively targeted to dopamine hub synapses. We identify several canonical proteins of dopamine synapses as highly enriched (DAT, VMAT2, Th for instance). D1Rs are not enriched, which corresponds with the presence of D1R on both synaptic and extra-synaptic compartments[30,65]. The D2R protein escaped MS/MS detection, which is likely due to the instability of the receptor in SDS denaturation buffer. With the validation of six targets using immunofluorescence assay, we show that our screen quality is high even for proteins with a low enrichment factor-like Cpne7 (Fig. 4). Some proteins are likely to populate presynapses of the hub partners (Cpne7, Mint-1/Apba1, Cadps2, Cadm2/SynCAM 2, Mgll) while Stx4 is more likely a postsynaptic protein based on the expression profile of the mRNA[35] and in accordance with the previous publications[41,44,46,47,49]. The immunofluorescence analysis of Stx4 expression confirmed a postsynaptic location distant to dopaminergic varicosity and most likely at glutamatergic spines that represent around 45% of hubs (Figs. 4–6). Cpne7, Mint-1/Apba1, Cadps2 and Stx4 point to specific membrane trafficking features at dopamine hub synapses[37,41,43,47].

The cross-analysis of our screen with single-cell RNA sequencing data allowed us to spot the synaptic adhesion protein SynCAM 2/Cadm2 (Fig. 3 and Supplementary Fig. 3) as a potentially important player of adhesion at dopaminergic varicosities[45,66,67]. The immunofluorescence data confirm the strong expression of SynCAM 2 at dopaminergic varicosities (Figs. 4e, 6a, b, g). SynCAM 2 was also reported to label axons[68]. Therefore, we propose that SynCAM 2 is part of an axonal adhesion complex responsible for the formation of dopaminergic synapses and hub synapses with SPNs. SynCAM 2 is thought to engage in heterophilic interactions with SynCAM 1 or 4[45,46]. Interestingly, SynCAM 1 is involved in cocaine-induced synaptic plasticity in the striatum[69] and SynCAM 2 is implicated in food intake and energy balance[70], two phenomena directly related to the integrity of the dopaminergic system[71,72]. Besides, SynCAM 1 is thought to be preferentially acting at the postsynapses to induce presynaptic adhesion[73,74]. Hence, SynCAM 1 and 2 are strong candidates to mediate adhesion through heterophilic interaction at dopamine synapses, in addition to their roles at other types of synapses.

A previous contribution suggested that adhesion at dopaminergic synapses occurs through neuroligin 2 (Nlgn2)[75]. Even though we detected all four neuroligins and all three neurexins in our proteomic analysis, none of them displayed a specific enrichment through DA-FASS. Also, Nlgn2 mRNA is not enriched in SPNs[35]. Thus, Nlgn2 did not appear to us as a putative major player in dopamine hub synapses compared to SynCAM 2[35]. Nlgn2 action is preeminent at inhibitory synapses[76,77]. To our knowledge, there is no record for the expression of Nlgn2 on spines or terminals of excitatory synapses which would be required in the context of dopamine hub synapses with excitatory inputs[77]. One possible explanation could be that Nlgn2 plays a role in the association with inhibitory synapses in the context of synaptic hubs. Also, some findings suggest a direct inhibitory function of dopamine projection on SPNs through GABA signalling[78,79]. Nlgn2 may have a function

related to this inhibitory phenotype. Finally, a combination of synaptic adhesion molecules is certainly involved and further investigations will be important to clarify the complete machinery responsible for dopaminergic hub synapse formation and maintenance[80].

**Cellular organization of dopaminergic projections to the striatum.** The nature of dopaminergic synaptic structures is the topic of a long-standing debate. Previous anatomical investigations identified that the distribution of dopamine varicosities in the neuropil is biased toward proximity to glutamatergic or GABAergic synapses, but only a minority were shown to make synapses with a target structure in the striatum[4,5,8]. However, other authors reported a frequent occurrence of symmetrical synaptic contacts of dopaminergic thin axonal portions with SPNs spines or dendritic shafts[9,81–83]. Our current dataset strongly advocates for specific and frequent adhesion of dopaminergic axonal varicosities with target structures (Figs. 1h–l and 2). Indeed, around 50% of our EGFP+ varicosities displayed apposed D1R, while ~80% displayed D2R labelling (Fig. 2). This is in accordance with SPNs being the main target of dopamine terminals in the striatum, with roughly half of the SPNs expressing D1 receptors, while the D2 receptor is expressed by the other half[31,84] as well as by dopaminergic and other presynapses[30,85,86].

Moreover, our data reveal that adhesion at dopaminergic varicosity extends to synaptic hubs with glutamatergic and GABAergic synapses. We find that around a third of dopaminergic varicosities make hub synapses with putative corticostriatal VGLUT1 synapses, around 12% associate with putative thalamo-striatal VGLUT2 synapses, and more than a quarter is associated with putative VIAAT inhibitory synapses (Figs. 5 and 6h). Additionally, around 14% are also contacted by cholinergic inputs. Conversely, 41% of VGLUT1 striatal synaptosomes are contacted by a Th+ input ($32/(32 + 45) \times 100 = 41\%$; Fig. 7b, j). VGLUT1 and VGLUT2 synaptosomes display little to no association when probed from a striatal sorting from VGLUT1-venus mice (Supplementary Fig. 5f, g). Also, little documentation exists supporting a GABAergic innervation on spines of SPNs[8,31]. Hence, little overlap may exist between those hub associations, and up to around 80% of dopaminergic varicosities may adhere to hub synapses rather than conventional bipartite symmetric synapses. According to the literature, cholinergic inputs to these hubs may target dopaminergic varicosities[55,57]. Further investigations will be required to characterize this in detail. As we show that most of the cytosolic content of dopaminergic axons is engulfed in synaptosomes (Fig. 1b–d), we propose that most dopaminergic presynapses are associated with hub synapses, while a small minority is not bound to any target cell in the tissue. Such a high occurrence of synaptic hubs may explain previous observations that striatal dopaminergic synaptosomes sediment faster than other synaptosomes in a linear sucrose gradient[87]. At the ultrastructural level, synaptic hubs are composed of electron-dense terminals containing many SV profiles associated with clear varicosities much less populated with SVs (Fig. 1f, h). This is in accordance with previous work stating that dopaminergic terminals are less populated with SVs and appear less dense to electrons[5,81,83].

Further investigations will be necessary to unravel whether synaptic hub formation is a structural invariant common to all sub-divisions of the striatum and whether the proportion of dopamine hub synapses of different kinds can vary depending on subregions and/or physiological states. The existence of hub synapses in other structures and modulatory inputs is an open question.

**Dopaminergic input to cortico-striatal synapses correlates with increased synaptic markers**. Beyond showing the existence of dopamine hub synapses, we identified that the binding of Th varicosities to cortico-striatal synapses correlates with an increase in VGLUT1, Bassoon, Homer1c, and Synaptopodin and a modest decrease in PSD-95 (Fig. 7). We found that nearly all dopaminergic terminals seem to adhere to a postsynaptic element populated with cognate receptors. Yet, around 2/3 of the dopaminergic varicosities are thought to be "silent" at a given time in the striatum and do not contain the full active zone molecular complement[12,13,88]. In synaptosomes, we found less than 20% of Th varicosities containing Bassoon, a percentage increased to 40% when Th varicosities are involved in dopamine hub synapses. We also found 70% of Th synaptosomes positive for Rim1. The precise combination of active zone scaffolds necessary and sufficient for dopamine release at hub synapses remains to be established. Similarly, it remains unclear whether dopamine release is required to induce the molecular potentiation we describe or whether trans-synaptic signalling of adhesion complexes is sufficient[10]. It will be important to characterize the function of newly identified proteins such a Syntaxin-4 or SynCAM 2 in the differentiation process.

The discovery of a molecular differentiation at synaptic hubs provides a unique ex vivo paradigm to study the complex interactions of receptors—through signalling crosstalk or heteromeric interactions—identified in the past decades[14–17]. Therefore, the question of the co-recruitment of glutamate or GABA receptors with dopamine receptors at synaptic hubs is raised and the plasticity of this recruitment upon reward-based processing and in dopamine-related pathological states remains to be established. Beyond, several metabotropic receptors (to adenosine, cannabinoid, glutamate and acetylcholine) are also important players in the striatal integration of cortical and thalamic inputs. The increase in Homer1c suggests a potential involvement of metabotropic glutamate receptors in the differentiation process[60]. Also, both Homer1 and Synaptopodin were shown to be involved in calcium signalling regulation, a point of interest for future investigations[60,63]. Downstream targets of signalling such as ionic channels may also take part in the critical scaffolds at play[31,89].

Altogether, our work paves the way for a better understanding of dopaminergic synaptic transmission in physiology and pathology[90]. Future developments will allow a more thorough multi-omics[91] as proposed recently with other techniques[92,93]. More generally, results from our study and the work of Apóstolo and colleagues[94] on mossy fibre terminals of the hippocampus show that FASS synaptomics is a powerful workflow for exploring projection-specific synaptomes[18,95].

## Methods

**Animals**. We used a transgenic mouse line expressing *cre* recombinase under the control of the dopamine transporter (DAT-*cre*+; MGI:3770172, RRID: MGI:3770172)[23], WT C57BL/6 N littermates, as well as a VGLUT1^venus knock-in mouse line (MGI Cat# 5297706, RRID:MGI:5297706)[58]. Mice were maintained in C57BL/6 N background and housed in 12/12 LD with ad libitum feeding, 50–70% humidity, and 18–22 °C ambient temperature. Every effort was made to minimize the number of animals used and their suffering. The experimental design and all procedures were in accordance with the European guide for the care and use of laboratory animals and approved by the ethics committee of Bordeaux University (CE50) and the French Ministry of Research under the APAFIS n° 8944 and #21132.

**AAV Vector and stereotaxic injection**. Stereotaxic injections were performed in heterozygous DAT-*cre*+ and wild-type (WT) mice of either sex at 8 to 11 weeks of age[96]. An Adeno-Associated Virus (AAV) containing an inverted sequence of EGFP (AAV1 pCAG-FLEX-EGFP-WPRE, University of Pensylvania)[22] or mNeongreen (AAV1 pCAG-FLEX-mNeongreen-WPRE)[97] coding gene flanked by loxP-sites was injected into DAT-*cre*+ mice (Fig. 1 Panel 1). Saline injected littermates were used as autofluorescence controls. The stereotaxic injections were

performed in Isoflurane-anesthetized mice using a 10 µl NanoFil syringe and a 35 G beveled NanoFil needle (World Precision Instruments). Injection coordinates for the Substantia Nigra pars compacta (SNc) were anterior/posterior (A/P)− 3.6 mm, lateral (L)± 1.3 mm, dorsal/ventral (D/V)− 4.2 mm. Injection coordinates for the Ventral Tegmental (VTA) were with a 12° angle A/P− 2.9 mm, L± 1.6 mm; D/V− 4.6 mm. A/P and L coordinates are given with respect to the *bregma*, whereas D/V coordinates are given with respect to the brain surface (Fig. 1 Panel 1). The animals were euthanized after 28–35 days at the maximal viral EGFP/mNeongreen expression. For each fluorescence-activated synaptosome sorting (FASS) experiment, three to six DAT-*cre*+ mice and one WT mouse were used independently of their sex.

**Subcellular fractionation of synaptosomes**. The preparation of synaptosomes was adapted from a previously published protocol[25]. Briefly, animals were euthanized by cervical dislocation, decapitated and the head was immersed in liquid nitrogen for a few seconds. The striatum of WT and bright fluorescent parts of the striatum of DAT-*cre*+ mice were subsequently dissected under an epi-fluorescence stereomicroscope (Leica Microsystems, Germany, Fig. 1 Panel 2). Non-fluorescent control striata were dissected following anatomical borders. Samples were homogenized in 1.5 ml of ice-cold Isosmolar buffer (0.32 M sucrose, 4 mM HEPES pH7.4, protease inhibitor cocktail Set 3 EDTA-free (EMD Millipore Corp.)) using a 2 ml-glass-Teflon homogenizer with 12 strokes at 900 rpm. The homogenate (H) was centrifuged at $1000 \times g$ for 5 min at 4 °C in a benchtop microcentrifuge. The supernatant (S1) was separated from the pellet (P1) and centrifuged at $12,600 \times g$ for 8 min at 4 °C. The crude synaptosomes pellet (P2) was resuspended in 350 µL of isosmolar buffer and layered on a two-step ficoll density gradient (5 mL of 13% Ficoll and 5 mL of 7.5% Ficoll, both in 0.32 M sucrose, 4 mM HEPES). The gradient was centrifuged at $50,000 \times g$ for 1 h and 10 min at 4 °C (Thermo Sorvall WX Ultra 90 with a Th 641 rotor). The synaptosome fraction (SYN) was recovered at the 7.5 and 13% ficoll interface using a 0.5 ml syringe. For complete subcellular fractionation 200 µL of the P2 fraction was transferred to a 10 cm³ ice-cold glass/Teflon potter and quickly homogenized at full speed in 1.8 mL ultrapure water to create an osmotic shock. For synaptic vesicle fractionation the lysate was centrifuged at $25,000 \times g$ for 7 min at 4 °C. The lysate supernatant (LS1), was centrifuged at $200,000 \times g$ for 120 min at 4 °C. The supernatant LS2 was collected, concentrated to 50 µL and aliquoted. The pellet containing crude synaptic vesicles (LP2) was resuspended in 50 µL of ice-cold isosmolar buffer and aliquoted. The lysate pellet LP1 was centrifuged on the same discontinuous ficoll gradient as for synaptosomes (7.5–13%) at $60,000 \times g$ for 33 min at 4 °C and the fraction at the interface of the two gradients containing synaptic plasma membranes (SPM) was collected and aliquoted. For each complete subcellular fractionation independent experiment, three DAT-*cre*+ 15 weeks old mice of either sex were pooled.

**Fluorescence-activated synaptosome sorting (FASS) workflow**. After collection, sucrose/ficoll synaptosomes were stored on ice and sequentially diluted in ice-cold PBS with protease inhibitor as described above, and the lipophilic dye FM4-64 dye was added at 1 µg/ml to the solution to label all membrane particles (Fig. 1a). The FACSAria-II (BD Biosciences) was operated through BD's FACSDiva with the following settings: 70 µm Nozzle, sample shaking 300 rpm at 4 °C, FSC neutral density (ND) filter 0.5, 488 nm laser on, area scaling 1.18, window extension 0.5, sort precision 0–16–0, FSC (340 V), SSC (488/10 nm, 365 V), FITC (EGFP/mNeongreen) (530/30 nm, 700 V), PerCP (FM4-64) (675/20 nm, 700 V). Thresholding on FM4-64 was set with a detection threshold at 800. Samples were analyzed and sorted at rates of 18,000–23,000 events/s and a flow rate of 3. Control synaptosome samples (SYN) coming from three different genetic labelling strategies (DAT-Cre-EGFP or DAT-Cre-mNeonGreen or VGLUT1^venus) were collected using the "singlet" gate, and FASS synaptosomes were sorted against the "EGFP+" or "mNeongreen+" or "VENUS+" sub-gate of the "singlet" gate, sequentially (Figs. 1, 7, Supplementary Figs. 1 and 5). After sorting, samples were either centrifuged on coverslips of 12 mm diameter coated with gelatin (1% Gelatin, 1% Chromium potassium sulfate for details see[21]; $5 \times 10^5$ synaptosomes per coverslip at $6800 \times g$ for 34 min at 4 °C Beckman J-26XP with a JS 5.3 rotor) or filtered on 0.1 µm Durapore hydrophilic PVDF membranes (Merck-Milipore). Filters were flash-frozen and stored at −80 °C until use. Coverslips were then further treated and analyzed either for immunofluorescence imaging or for electron microscopy while filtered samples underwent WES or mass spectrometry analysis (Fig. 1a).

**Simple Western™ immunoblot**. Proteins collected on filters were solubilized in 70 µl of SDS-PAGE loading buffer at 70 °C for 3 min and cooled down on ice. Conventional gradient synaptosomes total proteins were titrated using the Bradford assay. FASS samples were titrated using silver staining performed on SDS-PAGE gels against a standard curve of gradient synaptosomes. Detection proteins of interest were determined using an automated capillary electrophoresis-based immunoblot to separate, identify and quantify a protein of interest (WES, ProteinSimple, USA). Reagents (Dithiothreitol, DTT; Fluorescent 5× Master Mix, Biotinylated Ladder) were prepared according to the manufacturer's protocol. Samples were diluted with 0.1× Sample Buffer and mixed with 5× Master Mix (4 to 1) to obtain 50 ng/mL and finally denatured 5 min at 70 °C. Primary antibodies were diluted to their tested optimal concentration and Luminol-Peroxide (1 to 1)

mix was prepared. The plate was filled following the protocol scheme (5 μL of Biotinylated Ladder, 5 μL of Samples, 10 μL of Wes Antibody Diluent, 10 μL of Primary Antibody, 10 μL of Streptavidin-HRP, 10 μL of Secondary Antibody and 15 μL of Luminol-Peroxide Mix). Simple Western™ standard immunodetection protocol was run (separation matrix loading: 200 s, stacking matrix loading: 15 s, sample loading: 9 s, separation: 25 min at 375 V, antibody diluent: 5 min, primary antibody: 30 min, secondary antibody: 30 min, detection: high dynamic range). Capillary chemiluminescent images captured through a charge-coupled device camera were analyzed by the manufacturer's Compass software. Briefly, the protein peak area under the curve (AUC) was fitted using a Gaussian distribution. The fitted protein AUC is expressed either as a ratio to the fitted AUC H fraction for each WES (Fig. 1) or as fitted protein AUC (Fig. 7).

**Immunofluorescence**. Synaptosomes on coverslips were fixed (4% Paraformaldehyde, 4% sucrose, 1× PBS) for 10 min at room temperature, washed three times with PBS for 5 min and then stored at 4 °C until use. Synaptosomes were blocked and permeabilized with PGT buffer (PBS, 2 g/L gelatin, 0.25% Triton X-100 and when needed 5% normal goat serum) and subsequently incubated with primary antibodies in PGT buffer (1 h at room temperature), washed three times with PGT and incubated with secondary antibodies in PGT (1 h at room temperature). Three final washes with PGT buffer were performed prior to a washing step in 1× PBS and a final rinse in ultrapure water. Coverslips were mounted on glass slides with Fluoromount-G mounting solution (Sigma) and stored at 4 °C until observation.

**Antibodies**. All antibodies used and their dilution are reported in Supplementary Table 1.

**Proteomics**
*Sample preparation and protein digestion*. Triplicates of 35*10^6 DA-FASS synaptosomes were accumulated for proteomic analyses and were compared to triplicates of 35*10^6 SYN synaptosomes. Synaptosomes counts were obtained through the cytometer software BD FACSDiva v9.0.1. Both samples were treated in parallel at all steps. Protein samples were solubilized in Laemmli buffer. A small part of each triplicate was analyzed by silver staining using SilverXpress^R staining kit (Invitrogen, Cat#LC6100). Protein content was normalized across triplicates to 140 ng (least concentrated sample of the series) and ran onto SDS-PAGE (Sodium Dodecyl Sulfate-Poly Acrilamide Gel Ellectrophoresis) for a short separation. After colloidal blue staining, each lane was cut in 2 bands, subsequently cut in 1 × 1 mm gel pieces. Gel pieces were unstained in 25 mM ammonium bicarbonate 50% Acetonitrile (ACN), rinsed twice in ultrapure water and shrunk in ACN for 10 min. After ACN removal, gel pieces were dried at room temperature, covered with the trypsin solution (10 ng/μl in 50 mM NH₄HCO₃), rehydrated at 4 °C for 10 min, and finally incubated overnight at 37 °C. Samples were then incubated for 15 min in 50 mM NH₄HCO₃ at room temperature with rotary shaking. The supernatant was collected, and an H₂O/ACN/HCOOH (47.5:47.5:5) extraction solution was added to gel slices for 15 min. The extraction step was repeated twice. Supernatants were pooled and dried in a vacuum centrifuge. Digests were finally solubilized in 0.1% HCOOH.

*nLC-MS/MS analysis and label-free quantitative data analysis*. The peptide mixture was analyzed on a Ultimate 3000 nanoLC system (Dionex, Amsterdam, The Netherlands) coupled with an Electrospray Orbitrap Fusion™ Lumos™ Tribrid™ Mass Spectrometer (Thermo Fisher Scientific, San Jose, CA). Ten microliters of peptide digests were loaded onto a 300 μm-inner diameter × 5 mm C₁₈ PepMap™ trap column (LC Packings) at a flow rate of 10 μL/min. The peptides were eluted from the trap column onto an analytical 75 mm id × 50 cm C18 Pep-Map column (LC Packings) with a 4–40% linear gradient of solvent B in 105 min (solvent A was 0.1 % formic acid and solvent B was 0.1% formic acid in 80% ACN). The separation flow rate was set at 300 nL/min. The mass spectrometer operated in positive ion mode at a 1.8 kV needle voltage. Data were acquired using Xcalibur 4.3 software in a data-dependent mode. MS scans (m/z 375–1500) were recorded in the Orbitrap at a resolution of R = 120 000 (@ m/z 200) and an AGC target of 4 × 10⁵ ions collected within 50 ms. Dynamic exclusion was set to 60 s and top speed fragmentation in HCD mode was performed over a 3 s cycle. MS/MS scans were collected in the Orbitrap with a resolution of 30 000 and a maximum fill time of 54 ms. Only +2 to +7 charged ions were selected for fragmentation. Other settings were as follows: no sheath nor auxiliary gas flow, heated capillary temperature, 275 °C; normalized HCD collision energy of 30%, isolation width of 1.6 m/z, AGC target of 5 × 10⁴ and normalized AGC target of 100%. Advanced Peak Detection was activated. Monoisotopic precursor selection (MIPS) was set to Peptide and an intensity threshold was set to 2.5 × 10⁴.

*Database search and results processing*. Data were searched by SEQUEST through Proteome Discoverer 2.5 (Thermo Fisher Scientific Inc.) against the *Mus musculus* SwissProt protein database (v2021-02-04; 17,050 entries) added with the green fluorescent reporter (mNeonGreen). Spectra from peptides higher than 5000 Daltons (Da) or lower than 350 Da were rejected. Precursor Detector node was included. Search parameters were as follows: mass accuracy of the monoisotopic

peptide precursor and peptide fragments was set to 10ppm and 0.02 Da, respectively. Only b- and y-ions were considered for mass calculation. Oxidation of methionines (+16 Da), phosphorylation of serines, threonines and tyrosines (+79), methionine loss (−131 Da), methionine loss with acetylation (−89 Da) and protein N-terminal acetylation (+42 Da) were considered as variable modifications while carbamidomethylation of cysteines (+57 Da) was considered as a fixed modification. Two missed trypsin cleavages were allowed. Peptide validation was performed using Percolator algorithm[98] and only "high confidence" peptides were retained corresponding to a 1% false positive rate at the peptide level. Peaks were detected and integrated using the Minora algorithm embedded in Proteome Discoverer. Proteins were quantified based on unique and razor peptides intensities. Normalization was performed based on the total protein amount. Protein ratios were calculated as the median of all possible pairwise peptide ratios. Background Based pairwise t-test was used to calculate p-values adjusted using Benjamini-Hochberg correction for the false discovery rate. This method assumes that most protein abundances usually do not change in response to any stimulus and automatically determines the range of protein ratios that are essentially constant between conditions and then tests each protein ratio against the median and variance derived from this background population. Quantitative data were considered for proteins quantified by a minimum of two peptides.

The mass spectrometry proteomics data have been deposited to the ProteomeXchange Consortium via the PRIDE partner repository with the dataset identifier PXD027534[99].

*Meta-Analysis with other databases*. Meta-analysis was carried out using databases from the mouse brain proteome[32], SynGO[33] and DropViz[35]. Volcano plots and heatmaps were created using python based bioinfokit[100].

**Epifluorescence microscopy and image processing**. Immuno-stained synaptosomes were imaged using either a Nikon Eclipse NiU (with a ×40/NA 0.75 dry objective equipped with a sCMOS ANDOR Zyla 5.5 camera), a Leica DMI8 epifluorescence microscope (with a ×63/NA 1.4 oil immersion objective equipped with a sCMOS Hamamatsu FLASH 4.0v2 camera) or a Leica DM5000 epifluorescence microscope (with a ×40/NA 1.25 immersion objective equipped with a sCMOS Hamamatsu FLASH 4.0 camera) and Metamorph software. Ten to twenty frames were chosen randomly on each coverslip and imaged.

Correlation of synaptosomes' labelling is automated by a homemade macro-command, using the ImageJ software[101] (SynaptosomesMacro_Randomization, version 10, with functional description and code that can be accessed here https://github.com/fabricecordelieres/IJ-Toolset_SynaptosomesMacro). The workflow is composed of three steps. First, the images are pre-processed. The original images, trans typed to 32-bits, are centred and reduced: their respective average intensity is subtracted and division by their standard deviation is performed. It is assumed that both signals lay close one from the other: both images are therefore combined into one to serve for synaptosomes' detection. On each pixel, the maximum signal from both channels is retained to produce a new image, subjected to both median filtering and gaussian blurring (3 pixels radius). Each potential synaptosome now appears as a bell-shaped blob, which centre might be determined using a local maximum detection (tolerance to noise: 3). Second, the detections are reviewed and user-validated. Part of the original images is cropped around the local maxima and displayed to the user as a mosaic. Each thumbnail is displayed on a clickable frame, allowing the user to include or reject a signal detection from the analysis. Criteria of rejection included: the presence of competing particles in the quantification area, bad focus on the particle, proximity of the image border preventing proper quantification. Finally, data is extracted, exported, and displayed. A circular region is positioned over the centre of the thumbnail. The centroid's coordinates are retrieved and logged. From the two sets of coordinates (one per channel), the inter-signal distance is computed and stored in μm. Signal quantification is performed by placing a circular region of interest (24 pixels radius) around the centroid, and measuring the integrated intensity. A measurement of the local background is performed through a doughnut-shaped ROI surrounding the measurement circle. All values are logged for both channels, for all retained structures and reported in a "_Pooled_CytoFile.csv" file. Further analysis was performed using the FlowJo and GraphPad PRISM softwares. xy-plots of integrated intensity values are displayed with a quadrant analysis of single or double signal detections. Quadrant gate positions were defined from raw images using the distribution of fluorescence intensities in the ROI. In the range of fluorescence intensities between the background peak and the next peak, we determined the quadrant borders at the values where the ROI start to show a clear punctum of the signal. For all analyses, randomly chosen particles were displayed in a gallery to give an overview of the population analyzed.

For association analysis, a randomization plugin was developed and integrated into the "SynaptosomesMacro_Randomization". The source code is available here: https://github.com/flevet/RandomizerColocalization, while the compiled version can be found here: https://github.com/fabricecordelieres/IJ-Toolset_SynaptosomesMacro/blob/master/Plugins/RandomizerColocalization_.class. Two separated particles were considered associated if d < 2 μm, with d being the Euclidian distance between their centroid. To statistically determine if these associations were significant or happening by chance, we performed randomization tests. Previously stored particles centroids coordinates were retrieved. For each

colour channel, we fixed the position of its particle while randomizing all the ones of the other channel using 10,000 Monte-Carlo simulations. Since there is no underlying structure, the probability of having a particle at a certain position is identical for the whole image space. Consequently, randomization was performed by generating a complete spatial random distribution having the same number of points as the number of particles of the channel being randomized. Associations between 2 markers were then computed as explained above. The final random association values reported were defined as the mean of 10,000 randomizations. All randomization values expressed in percentages are logged for both channels, for all retained structures and reported in a "_Pooled_RandomizationResults.csv" file.

**Stimulated emission depletion (STED) microscopy**. Images were acquired using a Leica TCS SP8/STED3X microscope equipped with a HC PL APO 93×/1.30 GLYC motCORR – STED WHITE objective and Leica Application Suite X (LAS X) software. We used depletion laser lines at 592 nm for Alexa488 and 775 nm for Alexa594 or ATTO647n fluorophores. A 25% 3D-STED effect was applied to increase Z resolution. Metrology measurements were regularly performed using fluorescent beads to test proper laser alignment. Less than 2 pixels (pixel size = 20 nm) shift between channels was measured.

**Electron microscopy**. Synaptosomes for transmission electron microscopy were fixed right after centrifugation on coverslips with a 1% Glutaraldehyde and 2% PFA (Electron Microscopy Sciences) in 1× PBS solution and kept at 4 °C until further treatment. They were then washed with PB and post-fixed in 1% osmium tetroxide and 1% $K_3Fe(CN)6$ in PB for 2 h on ice in the dark. Washed in $H_2O$ and dehydrated in an ascending series of ethanol dilutions (10 min in 50% ethanol, 10 min in 70% ethanol, twice 15 min in 95% ethanol, twice 20 min in absolute ethanol). After absolute ethanol, coverslips were lifted into Epon 812 resin (Electron Microscopy Sciences) and 50% ethanol for 2 h at room temperature and then left in pure resin overnight at 4 °C. Coverslips were then placed on microscope slides, embedded with capsules filled with pure resin and polymerized at 60 °C for 48 h. The resin block was then trimmed with razor blades. Sections, 65 nm thick, were then cut using a diamond knife Ultra 35° (Diatome) with an ultra-microtome (Leica UC7) and collected on 150 mesh copper grids (Electron Microscopy Sciences).

The sections were stained with UranyLess® (Chromalys and Deltamicroscopy). Samples were then observed with a Hitachi H7650 transmission electron microscope equipped with a Gatan Orius CCD camera. Synaptosomes were identified by their size (0.5–2 μm), their shape and the presence of intracellular compartments and organelles such as vesicles.

**Statistics and reproducibility**. Sorts' statistical analysis was performed using two-way mixed design (MD) ANOVA (Fig. 1f and Supplementary Fig. 5e).

WES' data statistical analysis was performed using Two-way ANOVA and Šídák's multiple comparison testing (Supplementary Table 2, Figs. 1d and 7c).

Proteomics data was tested using a Background Based pairwise t-test used to calculate p-values adjusted using Benjamini-Hochberg correction for the false discovery rate and is described in the proteomics section of the methods. Proteins with an abundance ratio above 1.5 or below 0.75 were considered different to the control providing that data displayed a statistically adjusted p-value lower than 0.05.

Statistical analyses for immunofluorescence's data were performed using two-way ANOVA (Figs. 2c–k, 4b–g, 5b–h, 7b and i, Supplementary Figs. 4a, 5e and 6b) and Šídák's multiple comparisons test. When comparing two populations, D'Agostino & Pearson test for normality followed by unpaired t-test when data were normally distributed or, otherwise, non-parametric Mann–Whitney testing (Fig. 7d–e–g; Supplementary Fig. 6c). When comparing cumulative frequencies, Kolmogorov–Smirnov test was used (Fig. 7d–h). Statistical analyses for distances' data were carried out after removal of outliers using ROUT's (Q = 1%) method (Th: distances analyzed n = 1206, Outliers n = 31; SynCAM 2: distances analyzed n = 979, Outliers n = 30; D2R: distances analyzed n = 187, Outliers n = 6; VAChT: distances analyzed n = 154, Outliers n = 3; VGLUT1: distances analyzed n = 199, Outliers n = 6; VIAAT: distances analyzed n = 367, Outliers n = 5; VGLUT2: distances analyzed n = 157, Outliers n = 7; Stx4: distances analyzed n = 903, Outliers n = 23; D1R: distances analyzed n = 258, Outliers n = 12), followed by D'Agostino & Pearson test for normality and non-parametric testing using Kruskal–Wallis' test with Dunn's multiple comparison (Fig. 6a).

No statistical method was used to predetermine sample size. Synaptosomes detected during image analysis were excluded when overlapping biological material prevented an accurate quantification or when the focus was not good enough for accurate quantification. When possible, experimental results were confirmed using a different method (WES immunoblots, different sort criteria like VGLUT1$^{venus}$ for instance). Each replication attempt has been successful in reproducing the data. The experiments were not randomized. The Investigators were not blinded to allocation during experiments and outcome assessment.

**Reporting summary**. Further information on research design is available in the Nature Research Reporting Summary linked to this article.

## Data availability

All data supporting the findings of this study are provided within the paper and its Supplementary Information. DA-FASS Proteomic dataset is publicly available at the ProteomeXchange Consortium http://proteomecentral.proteomexchange.org/cgi/GetDataset via the PRIDE partner repository with the dataset identifier PXD027534. The mouse brain proteome is available at http://www.mousebrainproteome.com/. SynGO database is available at https://syngoportal.org/index.html. DropViz database is available at http://dropviz.org/. KEGG database is available at https://www.genome.jp/kegg/pathway.html. Source data are provided with this paper as Source data file. The WES, FACS and immunofluorescence data generated in this study have been deposited in the Zenodo database (https://doi.org/10.5281/zenodo.6482952). Because of their size, the raw microscopy images underlying the results will be made available upon request to the corresponding authors. Requests will be answered within a week. Source data are provided with this paper.

## Code availability

The macro code to perform automated correlation of synaptosomes' labelling and association analysis are available on zenodo at the following link: https://doi.org/10.5281/zenodo.6483605.

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

## Acknowledgements

Our work benefited from the excellent technical support from the central facilities at Bordeaux university: Bordeaux Imaging Center (CNRS UMS 3420, INSERM US4); Biochemistry and Biophysics of Proteins; Flow cytometry UB'FACSility (CNRS UMS 3427, INSERM US5); animal care & breeding; Genotyping; proteomics. We thank Niels Christian Danbolt for the kind gift of antibodies. We are grateful to François Georges, Philippe Vernier, André Callas, Nicolas Heck and Peter Vanhoutte for fruitful discussions. Julie Angibaud, Lou Bouit, Elisabeth Normand, Hajer El Oussini, Guillaume Dabee and Melissa Deshors provided an excellent technical support. This study benefited from the the Agence Nationale de la Recherche consortium fundings (IDEX Bordeaux ANR-10-IDEX-03-02; Labex BRAIN ANR-10-LABX-43 BRAIN; France Bio Imaging ANR-10-INBS-04) as well as funding to P.T. and E.H. (ANR-10-LABX-43 BRAIN Dolipran), funding to P.T. (ANR-16-CE16-0022 SynLip) and funding to T.B. (National Institutes of Health R01 DA018928). E.H. received funding from Fondation pour la Recherche Médicale (ING20150532192) and E.H. and D.P. from ANR (DopamineHub-19-CE16-0003-01) and from Conseil Régional de Nouvelle Aquitaine (ParkSynGref). R.W. holds a fellowship from the French ministry of research.

## Author contributions

Experimental design: D.P., E.H., M.E.P., M.Pr., P.T., V.P.B.; Experimental work: E.H., M.E.P., M.Pr., M.F.A., P.L., V.P.B.; Technical contribution to experiments: C.M., F.P.C., F.L., M.F.A., M.Pe., R.W., S.C., S.L., V.D.P., V.P.; Software programing and data analysis: V.P.B., E.H., F.L., F.P.C., M.E.P., M.Pr., P.L., S.C.; Reagent sharing: T.B.; Funding: D.P., E.H., P.T.; Article writing: D.P., E.H., M.E.P., M.Pr., P.L., P.T., V.P.B.; Article editing: all authors.

## Competing interests

The authors declare no competing interests.
