## [Peer review file · Nature Communications]

REVIEWER COMMENTS

Reviewer #1 (Remarks to the Author):

This manuscript describes a methodological approach based on fluorescent tagging of dopaminergic neurons in mice. Fluorescence activated particle sorting was used to enrich synaptosomes (termed FASS) from dopaminergic terminals in striatum, and mass spectrometry used to determine protein identity and abundance. Extensive, follow-up validation using immunological probes and high resolution imaging was then carried out to characterize the proteome of dopaminergic synapses formed on striatal spiny neurons.

General comments:

The work expands on a previously published method to enrich synaptosomes using FACS/FASS. Technically, the work is carefully done and well carried out, with adequate secondary validation and quantitation. The work is inherently descriptive with no biological manipulations. The work will be a useful technical report for researchers studying striatal biology.

Major:

1. The manuscript is densely written and presented. While understandable, the quality of the writing could be improved.
2. As noted, the work is descriptive. As detailed in the final sections of the Results, and summarized on L525-526 – “To our knowledge this is the first time a physical interaction of dopamine axons with their target is shown to induce molecular differentiation.”, the authors infer causality, when only correlation is shown.

Minor:

L124-127 – should this be split into two sentences?

L187 – “spectrometry”

Fig 5 – why is PKC shown in red?

L687 – “Sybnaptopodin” – typo?

L709 – “Spots” – what does this mean.

L756 - PXD027534 95) – remove).

L769-795 – writing – L795 – reference to Fig 1C is presumably in error

L821 – is reference to Fig 1D intentional?

L1154 – plan/plane

Reviewer #2 (Remarks to the Author):

The work from Paget-Blanc and co-authors tackle an important and unresolved question in the field of synaptic biology. Namely, it ambitions to characterize the proteomic composition of dopaminergic synapses. While the proteomes of glutamatergic and gabaergic synapses are well characterized, those of modulatory synapses, such as dopaminergic ones, have not been defined so far. The authors take advantage of a technology developed (FASS + LC-MS/MS) by the group, which has great potential. Furthermore, it is obvious that the amount of effort and work presented in this manuscript is enormous and the overall technical rigour is high. Nonetheless, I found several important issues that should be addressed.

Major comments:

1. The fact that DA axons contact other synapses either at the pre-synaptic button or at the post-synaptic spine (these seem to be more common) is, to the best of my knowledge, not new. The authors actually reference some of the papers that report these observations (i.e. Freund 1984; Moss 2008 or Uchigashima 2016). Because of this, I am a bit surprised that the authors propose the existence of 'hub synapses' as something not previously described. These could very well reflect the known interaction of dopaminergic axons with other types of synapses (glutamatergic, GABAergic, cholinergic, etc), which are recovered in their experiments.

Similarly, the authors propose that the two presynaptic elements they observe co-localizing (i.e DA+Glut; DA+GABA; DA+Ach) will both contact a postsynaptic element. I have not found any prove or hint for that hypothesis in their manuscript. And, although this is presented as an hypothesis, in my opinion these should be removed as there is no reason why axo-axonic contacts could not be represented in their samples. In my opinion the most parsimonious explanation is that they will have a mixture of axo-axonic synapses and presynaptic boutons contacting dendritic spines.

I found very interesting that 31%+15% of DA boutons would colocalize with vGlut1/2; 25% with vIaAT and 14% with vAChT. This adds to 85% DA boutons contacting other synapses; implying that the vast majority of DA terminals impact other synapses. This, I believe, is less well known in the field, and would be rather novel. It might be very interesting to perform similar experiments with other modulatory synapses (i.e. serotonergic synapses; although I am not suggesting that this should be done in the context of the present manuscript), to see if this characteristic is unique to DA synapses or if it is a common trend among modulatory synapses.

Since most DA synapses isolated are likely attached to another synapse, each of the particles studies will have 2 pre-synaptic elements and 2 postsynaptic elements. This makes a bit more difficult to interpret where are the proteins found enriched in the MS data (57 proteins). Still it is plausible that they will be more strongly associated with the dopaminergic pre- or post-synaptic element of these 'hub synapses', but this potentially confounding fact should be discussed in more detail.

2. In Figure 3 comparing the overlap of their proteomic dataset with that of glutamatergic and gabaergic synapses should be done. The comparison with mouse brain proteome does not really add much, it is expected that this will be very high.

3. The finding of DA colocalizing with vGlut1/2 would contradict, at least in part, the findings made by Uchigashima et al (ref 70 in the manuscript). In this paper they claim that dopamine synapses are formed with GABAergic postsynaptic elements only. This is an interesting finding and should be discussed/highlighted.

4. I have not found in the manuscript if the ANOVA done to compare protein abundances between unsorted singlets and DA-FASS has been corrected for multiple testing. This is a standard procedure in proteomics and it should be done (if it hasn't). It might be that the authors end up with no proteins with a significant differential expression after correction for multiple testing. This is mainly because MS data is poorly suited to be analysed with and ANOVA +FDR. If this is the case I would strongly recommend the authors to use the MSqROB package for statistical analysis of label-free proteomics data. In our experience this works much (much) better than using ANOVA. MSqROB will include a correction for multiple testing (FDR). In our experience the final list of significantly different proteins does not vary when using ANOVA or MSqROB.

5. I could not find in the methods section how are antibodies incorporated to synaptosome preparations prior to FASS. Also, based on Figure's 2 Legend, an antibody against EGFP is used for FASS. If this is correct, it is a bit surprising, as one would expect the intrinsic fluorescence of EGFP to be used for sorting. This should be clarified in the methods. This would also be relevant when sorting for EGFP and D1R or D2R (Fig 2), in what conditions are the antibodies added to synaptosomes prior to FASS? I do not think this is explained in the methods.

Minor points:

- The miniaturization of the centrifugation methods used to isolate synaptic and subsynaptic elements could be better explained in the methods, as these might be of use to other groups.
- It is my understanding that when the FASS sorted material is filtrated using 0,1um filters the material attached to the filter is what is later used for WB or MS; not the material that passes through the filter. It should be clarified how this is done.
- It might be interesting to discuss why in LP2 (Fig 1C) there is some Th while no EGFP. There might be some TH associated to SVs or other membranous structures.
- In Fig 1A3 (or the figure legend) it would help to clarify that B correspond with the synaptosomal fraction (B=SYN in Fig1C)
- Fig1AE in x axis legends are not well displayed
- Lane 136 / Lane 137: I think the text 'a significant proportion' does not refer to any statistical analysis. The term 'significant' is normally used in the context of a statistical analysis. More importantly in several instances throughout the text vague language is used (as in here). This should be avoided, providing numeric values whenever possible.
- Lane 146. It is unclear in what sense the word 'specific' is used in this context. Why specific? Consider re-writing in a more precise style.
- Could help the general audience to better define what are unsorted singlets.
- Lanes 158/159 and 162, please add reference to figures. Data in lane 162 is refered to Fig S2, no?
- Lane 171: Clarify how this 55% percent is calculated. I understand it is from $32/(26+32) \times 100$.

- L182: 'nearly all': unprecise, please try to add numeric values to these statements when available.
- Typo: L187: 'spectrometric' should be 'spectrometry'
- Would be interesting to discuss why dopamine receptors are not found enriched in the MS analysis. Actually it is my understanding that only D1 is actually identified among the 26700+ proteins found by MS.
- L237: shouldn't CIN cells be in figure 4A and S3.
- L263: SynCAM2 is used together with DAT, while all other proteins are analysed with TH, why? Same thing in L331: VACHT is analysed with TH and the other transporters with EGFP, why?
- L709: typo, 'spots' should be 'bands'
- Adding a table in supplementary with all details regarding the large number of antibodies used could be quite useful.

Reviewer #3 (Remarks to the Author):

The striatum is a brain region that is involved in cognitive, motor and emotional processes. Medium spiny neurons, the principal neurons of the striatum, receive inputs from glutamatergic, GABA-ergic, but also dopaminergic neurons. Despite the known modulatory effects of dopamine on glutamatergic and GABA-ergic synapses, as well as the physiological and medical importance of the dopaminergic pathway, there is still no consensus regarding the relative importance of synaptic and extrasynaptic dopamine release in the striatum,

The authors developed a cell fractionation and fluorescence sorting procedures to obtain striatal synaptosomal fraction where dopaminergic inputs are fluorescently labeled. This allowed them to characterize the molecular composition of the synapse types present in the striatum by immuno-labeling and mass spectrometry, as well as the relative localization of synaptic components. Combining this data, they provide evidence that glutamatergic and GABA-ergic synapses associate with dopaminergic presynaptic terminals to form a structure termed hub synapses. While dopaminergic afferents localized in the vicinity or making synaptic contacts with glutamatergic synapse have been observed by electron microscopy before, the characterization of these structures would make this manuscript suitable for Nature Communications. However the manuscript does not seem to be completely finished, there are many (mostly small) mistakes and the data is not always presented in a way that clearly leads to the conclusions. Here are the specific concerns:

1. Parts of the manuscript are difficult to find. Specifically, it should be clear which files contain supplementary tables. Currently, these files have cryptic names, one can guess which ones are Tables S3-S5 from their content. There are no references for Tables S1 and S2, but there are two additional tables that are not labeled.

2. This is a complex manuscript that uses lot of abbreviations. In its current form, the terminology used makes some parts of the manuscript difficult to read. Therefore I suggest that the terminology is improved, the abbreviations should be consistent and it is not advisable to use different terms that have the same meaning. For example:

(a) Please define “effector synapses”. It might be even better to use “glutamatergic and GABA-ergic synapses” instead.

(b) (Lines 124-6): Please explain differences between “DAT-EGFP”, “singlet EGFP” and “FASS EGFP”, a simple scheme might help.

(c) (Figure 1F): The use of “singlets” is inconsistent and confusing. The two bars on the left (a dark and the light green) are labeled “singlets” below the x-axis, however the three dark green bars are labeled “singlets” in the legend. Also, I suppose EGFP- and EGFP+ bars are also singlets (as opposed to aggregates), which makes all of the bars singlets, regardless of the labeling. Please clarify what the bars mean.

(d) Another example is “unsorted singlets” as opposed to “FASS-sorted synaptosomes” (line153), even though the latter are probably also singlets. The same for “Singlets” and “DA-FASS” on figures.

(e) Do DAT-EGFP, DA-FASS and EGFP+ FASS mean the same ? If yes, please use only one.

(f) DAT+/EGFP-, EGFP+/D1R- and EGFP+/D1R+ are examples of clear nomenclature. I suggest using this form for all other cases, for example DAT+/EGFP+ instead of DAT-EGFP

(g) There is a change from "DAT-EGFP labelled synaptosomes" to "DAT-EGFP labelled varicosities" (lines 166, 169). Please state whether synaptosomes and varicosities are the same.

(h) Labeling like "Th+/EGFP-" and "TH-/EGFP+" instead of "Th+" and "EGFP+" would make bar charts easier to understand (Figure 2C, ...)

3. Shouldn't the ratio of EGFP+ to EGFP- (the upper two quadrants to the lower two) be the same on Fig 2 A, E and I because the sample is the same. This ratio is much higher on panel A than on E and I.

4. Regarding the statistical analysis:

(a) At several places, a change in percentage of dots in a quadrant is reported in the manuscript, but no statistical significance is given (line 177, 293, ...).

(b) It is stated that two-way mixed design ANOVA was used and the results are shown in one of the unlabeled files (Figures like 2C, 2G, ...). However it is not clear what are the ANOVA factors (fixed treatment and random effect), because the sorting, EGFP and another factor (Th, D1R, ...) are present.

(c) Whenever the statistical significance is stated, what is shown to be different

5. It is confusing to find numbers in the text that are not shown on the corresponding figures. For example, I am aware that "27%" (line 323) is correct because it comes from "20% and "54%" (Figure 5F). However, making these obvious would help the reader

6. Given the proteins investigated in this section, it would be more precise to use "dopaminergic presynapses" and "all the major presynaptic partners" (lines 335-6). Similar issue on line 351

7. How are “co-localized”, “apposed” and “rather distant” defined (lines 367 and 369)? Quantification is needed for these statements in order to justify the model (lines 372-5). Does “either D1 or D2” come from the data presented here or elsewhere? It is not clear how is the sentence on line 365 related to the model.

8. There are several issues regarding the detection of Bassoon:

(a) Please state the p-value (lines 390, 397)

(b) The qualification “close to the active zone” is not supported by Figure S6A (line 393)

(c) It is not clear why Th-/VGLUT1+ is compared with Th+/VGLUT1- and which conclusions can be made from this (lines 392-4)

(d) Two way ANOVA analysis of his data might determine the interaction between VGLUT and Th

(e) STED data need to be statistically analyzed in order to justify the conclusions (lines 399-402). The possibility that the higher Bassoon signal in Th+/VGLUT1+ compared to Th-/VGLUT1+ is simply due to the contribution from those Th presynapses that contain Bassoon.

9. Statistical significance should be stated for results on lines 406-417)

10. Is it possible that dopaminergic presynapses do not make synaptic contacts to glutamatergic (or GABA-ergic) postsynapses as proposed for hub-synapses, but that dopaminergic and glutamatergic (and GABA-ergic) synapses do not share the postsynapse and are kept together by the direct association of the presynaptic terminals, like on Figure 1J?

11. It is not clear on which basis this assumption is made (line 504)

12. As noted above, the statements on lines 522-524 are not justified

13. The method for measuring distance between two labeled dots should be specified in more detail:

(a) Is the source code available?

(b) What features are shown by the “binary representation”?

(c) Please explain how was the wavelet filtering used

(d) Are “objects” and “particles” the same?

(e) What was the size of analyzed images? Like Figure 2D?

(f) How many objects were present on one image?

(g) Was the randomization done over the whole image or a portion of it?

(h) Isn't the probability of having a random particle at $< 2 \mu\text{m}$ to the fixed particle simply the fraction of the image area that is $< 2 \mu\text{m}$ to the fixed particle?

14. Considering that the extracellular region of SynCAM2 was labeled, why is the SynCAM2 label relatively far from the DAT label (Figure 6G)?

15. Synaptosomal preparation:

(a) The synaptosomal preparation used is termed “the classical sucrose fractionation” (line 103), while the ficoll preparation is described in the Methods.

(b) How was the number of synaptosomes determined (line 105)

16. Please explain “gelatinized coverslips (line 633)

17. (Line 119-120): "Soluble proteins" is not clear. What is "S2"? It cannot be said "most of the soluble protein ..." when only two proteins are checked (EGTA and Th).
18. (Line 132): Munc18 appears more than "slightly reduced" on Figure 1G
19. Please quantify the statements "significant proportion" (line 136) and "another example" (line 139)
 - (a) "Collapse aggregates" is not clear (line 145). Would there be a possibility that cell sorting induces aggregation?
20. What criterion was used to determine the intensity separation lines on dot plots (Figure 2A, B, D, F, ...)?
21. Please explain where is shown the "5-fold" increase (line 159)
22. Co-localization of what with anti-DAT (line 161)
23. The argument on lines 162-167 is confusing.
24. No justification is given for calling (EGFP-) D1R extra-synaptic (line 170)
25. Please quantify "closely or more distantly" (line 179)
26. Please specify "very high" (line 288)
27. It is not clear where do the statements about apposition (lines 294 and 301) come from
28. Please be more specific (lines 207, 306)

29. Considering that immuno-fluorescence signal centers are found for each protein separately, wouldn't "center of mass" be more appropriate than "barycenter" (spelled wrong) (line 359)
30. Figure 5D is not related to Th (line 380)
31. Rather "decreased" than "maintained" (line 384)
32. "Significant" would be more precise than "strong" (line 427)
33. References 4 and 5 are cited as supporting two opposite conclusions (line 489-91)
34. Figure 7J is not cited.
35. Please state the pixel size wherever distances are specified in pixels (lines 777, 818)
36. Which ".CSV file" (line 791)?
37. Figure 1C is not cited in the correct context (line 795)
38. Symbols "b" and "m" are not specified for Fig 1L (line 1154)
39. Please state N (line 1252). Also, where do "55%" and "45%" come from (line 1259)?
40. Where do "60%" and "40%" on Figure 7J come from?
41. There are several cases of spelling mistakes and inappropriate word use. For example: "aimed" (line 85), "decisively" (line 311), "challenge" (line 337), "co-sedimentation" (line 339), words missing (line 349), "each markers" (line 359), "reports for" (line 387), "harbour a cluster of Bassoon" (line 392), "until now on the other hand" (line 449), "engage into" (line 511), "are clear to electrons" line (512-3), "plan" (line 1154), "VGLUT2 thalamo" instead of "VGLUT1 thalamo" (line 1343).

We first wish to thank all 3 reviewers for their careful reading of our report, with very constructive comments. We are enthusiastic that the general opinion of the reviewers, points to the importance of the description of cellular and molecular features of dopaminergic synapses without further biological manipulations at this stage. We also acknowledge their perception of the amount of work needed to share this dataset with the community. We agree with the shared view of the reviewers that in its prior version our manuscript suffered from several flaws and inconsistencies in its writing and presentation. This reflects the accumulation of layers of work through a couple of editorial processes and we are truly sorry to not have spotted these flaws prior to submission. In the present form we addressed most if not all points raised by the reviewers as detailed below.

Reviewer #1 (Remarks to the Author):

This manuscript describes a methodological approach based on fluorescent tagging of dopaminergic neurons in mice. Fluorescence activated particle sorting was used to enrich synaptosomes (termed FASS) from dopaminergic terminals in striatum, and mass spectrometry used to determine protein identity and abundance. Extensive, follow-up validation using immunological probes and high resolution imaging was then carried out to characterize the proteome of dopaminergic synapses formed on striatal spiny neurons.

General comments:

The work expands on a previously published method to enrich synaptosomes using FACS/FASS. Technically, the work is carefully done and well carried out, with adequate secondary validation and quantitation. The work is inherently descriptive with no biological manipulations. The work will be a useful technical report for researchers studying striatal biology.

We are grateful to the reviewer for this general positive comment on our work.

Major:

1. The manuscript is densely written and presented. While understandable, the quality of the writing could be improved.

We worked on the text with the help of a naïve native english reader to improve the overall quality of the text.

2. As noted, the work is descriptive. As detailed in the final sections of the Results, and summarized on L525-526 – “To our knowledge this is the first time a physical interaction of dopamine axons with their target is shown to induce molecular differentiation.”, the authors infer causality, when only correlation is shown.

The point is well taken. The correlative nature of this finding is strengthened and the putative causality reduced in the text :

L539-541 – “Finally, we observed that innervation of glutamatergic synapses by dopaminergic varicosities correlates with a molecular strengthening of the whole synapse.”

*L634 – “**Dopaminergic input to cortico-striatal synapses correlates with with increased protein content.**”*

Beyond showing the existence of dopamine hub synapses, we identified that the binding of Th varicosities to cortico-striatal synapses correlates with an increase in VGLUT1, Bassoon, Homer1c and Synaptopodin and a modest decrease in PSD-95. To our knowledge this is the first time a physical interaction of dopamine axons with their target is shown to correlates with molecular differentiation.”

L646-649 – “Finally, it will be important to characterize the function of newly identified proteins such a Syntaxin-4 or SynCAM 2 in the differentiation process, and the ultrastructural correlate of the observed molecular plasticity.”

Minor:

L124-127 – should this be split into two sentences?

Corrected (Now L133-137)

L187 – “spectrometry”

Corrected (Now L218)

Fig 5 – why is PKC shown in red?

Proteins enriched in DA-FASS are labelled in red in all panels comparing our proteome data with KEGG database of synaptic pathways.

L687 – “Sybnaptopodin” – typo?

Corrected (Now Supplementary Table 1)

L709 – “Spots” – what does this mean.

Replaced by “Samples” (Now L804)

L756 - PXD027534 95) – remove).
Corrected (Now L855)

L769-795 – writing – L795 – reference to Fig 1C is presumably in error
Corrected (Now L902)

L821 – is reference to Fig 1D intentional?
Removed (Now L942)

L1154 – plan/plane
Corrected (Now L1277)

Reviewer #2 (Remarks to the Author):

The work from Paget-Blanc and co-authors tackle an important and unresolved question in the field of synaptic biology. Namely, it ambitions to characterize the proteomic composition of dopaminergic synapses. While the proteomes of glutamatergic and gabaergic synapses are well characterized, those of modulatory synapses, such as dopaminergic ones, have not been defined so far. The authors take advantage of a technology developed (FASS + LC-MS/MS) by the group, which has great potential. Furthermore, it is obvious that the amount of effort and work presented in this manuscript is enormous and the overall technical rigour is high. Nonetheless, I found several important issues that should be addressed.

We are grateful to the reviewer for this general positive statement regarding our study.

Major comments:

1. The fact that DA axons contact other synapses either at the pre-synaptic button or at the post-synaptic spine (these seem to be more common) is, to the best of my knowledge, not new. The authors actually reference some of the papers that report these observations (i.e. Freund 1984; Moss 2008 or Uchigashima 2016). Because of this, I am a bit surprised that the authors propose the existence of ‘hub synapses’ as something not previously described. These could very well reflect the known interaction of dopaminergic axons with other types of synapses (glutamatergic, GABAergic, cholinergic, etc), which are recovered in their experiments.

Similarly, the authors propose that the two presynaptic elements they observe co-localizing (i.e DA+Glut; DA+GABA; DA+ACh) will both contact a postsynaptic element. I have not found any prove or hint for that hypothesis in their manuscript. And, although this is presented as an hypothesis, in my opinion these should be removed as there is no reason why axo-axonic contacts could not be represented in their samples. In my opinion the most parsimonious explanation is that they will have a mixture of axo-axonic synapses and presynaptic boutons contacting dendritic spines.

I found very interesting that 31%+15% of DA boutons would colocalize with vGlut1/2; 25% with vLaaT and 14% with vAChT. This adds to 85% DA boutons contacting other synapses; implying that the vast majority of DA terminals impact other synapses. This, I believe, is less well known in the field, and would be rather novel. It might be very interesting to perform similar experiments with other modulatory synapses (i.e. serotonergic synapses; although I am not suggesting that this should be done in the context of the present manuscript), to see if this characteristic is unique to DA synapses or if it is a common trend among modulatory synapses.

Since most DA synapses isolated are likely attached to another synapse, each of the particles studies will have 2 pre-synaptic elements and 2 postsynaptic elements. This makes a bit more difficult to interpret where are the proteins found enriched in the MS data (57 proteins). Still it is plausible that they will be more strongly associated with the dopaminergic pre- or post-synaptic element of these ‘hub synapses’, but this potentially confounding fact should be discussed in more detail.

We thank the reviewer for this in-depth discussion of our findings regarding dopamine complex interaction with partners in the striatal neuropil. We will try to answer to each major comment within point 1:

Hub synapses vs a collection of axo-axonic and axospinous synapses.

We completely agree with the fact that the existence of dopamine synapses on spines or terminals had been documented previously and that this was thought to represent a minority of events among dopaminergic varicosities. An important feature of conventional axo-spinous excitatory synapses is to be resistant to homogenization and persist as synaptosome upon fractionation. The synaptosome being classically defined as the resealed terminal bound to the post-synaptic membrane and part of the post-synaptic density associated scaffolds (Whittaker., 1993 PMID: 7903689). The post-synaptic membrane is most of the time not resealed. We believe that beyond providing

an advanced fractionation of dopaminergic synaptosomes, an original proteomics dataset, a major finding of our work is the isolation of multipartite synaptosomes involving the dopaminergic varicosity. They form a mechanically resilient structure that can be isolated like regular synaptosomes are. In the 2 examples we show in Figure 1jk we clearly see 3 and 4 elements respectively bound together. This is more than isolating an axo-axonic or an axo-spinous. Our observation is then supported by the proteomics analysis displayed in Figures 3-5 and the epifluorescence analysis done in Figures 6 and 7. Additionally, for glutamatergic hub synapses we observe a co-enrichment of synaptopodin (a marker of the spine apparatus located in the spine base and neck; Figure 7) while we previously observed a steep depletion in glutamatergic synaptosomes with our FASS method (Biesemann et al., 2014 PMID: 24413018; Hafner et al., 2019 PMID: 31097639). We agree with the reviewer that having up to 80% of dopaminergic varicosities involved in multipartite associations suggests a much higher point-to-point targeting of dopaminergic inputs than previously anticipated. We therefore think that these multipartite synaptosomes represent meaningful biological structures of the striatal neuropil worth to be defined by a proper name and we chose "hub synapses" to qualify these structures of convergence of different inputs. We worked on the text to convey this message explicitly (L 96-98 / 163-168 / 416-419 / 439-443 / 524-526 / 535-541). Of course, much work is still needed to characterize the different classes of hub synapses and define much better this new concept but this is probably beyond the scope of this first study.

Whether all partners of synaptic hubs contact a post-synapse ?

In line with the answer above, our point is that when a varicosities labelled with a presynaptic marker (Th, DAT, VACht, VGLUT1 & -2, VIAAT) is found bound to another neuronal element this apposed element is a post-synapse. For instance, in Figure 1j we see that terminal b1 may synapse on the arrowhead, while b2 seem to synapse on b1. In that sense b1 may be the post-synapse of b2. In Figure 1k we see that the 3 boutons share a common post-synaptic element. A lot of work is still needed to understand how information flows between each partner of these scaffolds. This is on our agenda for next years.

Regarding the extension of the hub synapse concept to all classical neuromodulatory systems.

Indeed, we agree with the reviewer that it will be important in the coming years to explore other neuromodulatory systems to probe for the existence of similar hub synapses. We now have the whole workflow running the test this in the near future.

How selective of dopaminergic varicosities is the proteomics dataset?

Indeed, even though our intention was to target our proteomics to dopaminergic varicosities, our dataset is representative of hub synapses because at most only a very small minority of our synaptosomes represent free unbound dopamine varicosities. Unfortunately, we also cannot yet reach the highest degree of synaptosome sorting purity as we would ideally like and have to take into account the relatively high remaining random contamination. We think that the whole analysis of our dataset takes into account this degree of complexity. To that end, we performed a meta-analysis to compare our data with single cell RNAseq data of cells involved in the striatal neuropil (afferent, interneurons and efferent cells; Figures 3h, 4a, Supplementary Fig 3). Additionally, we show that many of the identified proteins that are kept through DA-FASS sorting belong to - or are common with other synapse types (Figure 5). We finally state this complexity in our current discussion (L 543-564). We also show that Cadm2/SynCAM 2 is a major component of hub synapses expressed at dopamine pre-synapses (but not only; see Figure 4 and Supplementary Fig 3) even though it is not specifically enriched in proteomics (Figure 3). Yet the 5- to 10-fold enrichment factor seen in FASS is sufficient to show that the proteins strongly enriched in our screen represent proteins relatively selective of hub synapses. This is depicted in our analysis of 5 of 6 targets validated in Figure 4. Finally, we provide additional epifluorescence analysis and STED microscopy data regarding Stx4 in the present version of our work. We show that Stx4 is present at 46% of dopaminergic synapses, but not within the pre-synaptic varicosity. Stx4 rather faces dopaminergic varicosities at an average distance of approximately 500nm. We reach the conclusion that Stx4 is most likely part of SPNs spines and recruited at hub synapses (Figure 6). Altogether, we think that our analysis strengthens the notion that dopamine hub synapses are built by the coordinated differentiation of several partners.

2. In Figure 3 comparing the overlap of their proteomic dataset with that of glutamatergic and gabaergic synapses should be done. The comparison with mouse brain proteome does not really add much, it is expected that this will be very high.

Indeed, this is an important point to address. We actually think that this is already done in the article. First, we compared our dataset to the SynGO database that was recently promoted by a large consortium of laboratories working on the topic of synapse cellular and molecular biology (Koopmans et al., 2019 PMID: 31171447; Figure 3). On the graph you find the number of genes in DA-FASS (green) versus SynGO curated gene list (gray) for each SynGO category. This database comprises mostly genes from either glutamatergic or GABAergic synapses. Additionally, we have made a systematic comparison of our dataset with the KEGG curated database for metabolic pathways. We compared our dataset with the pathways for dopaminergic synapse, presynaptic vesicle cycling, glutamatergic synapses, GABAergic synapses and cholinergic synapses (Figure 3, 5; details in Supplemental table 5). We do find a very significant overlap in both comparisons.

3. The finding of DA colocalizing with vGlut1/2 would contradict, at least in part, the findings made by Uchigashima et al (ref 70 in the manuscript). In this paper they claim that dopamine synapses are formed with GABAergic

postsynaptic elements only. This is an interesting finding and should be discussed/highlighted. *We agree with the reviewer, but additionally, our data may also be in line with those of Uchigashima if one considers the 25% of association with GABAergic synapses. We reinforced this discussion in the present text (L 580-594)*

4. I have not found in the manuscript if the ANOVA done to compare protein abundances between unsorted singlets and DA-FASS has been corrected for multiple testing. This is a standard procedure in proteomics and it should be done (if it hasn't). It might be that the authors end up with no proteins with a significant differential expression after correction for multiple testing. This is mainly because MS data is poorly suited to be analysed with and ANOVA +FDR. If this is the case I would strongly recommend the authors to use the MSqROB package for statistical analysis of label-free proteomics data. In our experience this works much (much) better than using ANOVA. MSqROB will include a correction for multiple testing (FDR). In our experience the final list of significantly different proteins does not vary when using ANOVA or MSqROB.

The point is well taken. We are very sorry that this was not properly described in the previous version of the manuscript and we thank the reviewer for pointing this. Actually, adjusted p-values corrected for multiple testing were already calculated in the original submission but the text was not precise enough. We modified the results and methods part in the present text (L 224 / 242-243 / 268 / 277 / 288 / 309 / 317 / 843-853)

5. I could not find in the methods section how are antibodies incorporated to synaptosome preparations prior to FASS. Also, based on Figure's 2 Legend, an antibody against EGFP is used for FASS. If this is correct, it is a bit surprising, as one would expect the intrinsic fluorescence of EGFP to be used for sorting. This should be clarified in the methods. This would also be relevant when sorting for EGFP and D1R or D2R (Fig 2), in what conditions are the antibodies added to synaptosomes prior to FASS? I do not think this is explained in the methods. *In fact, antibody stainings are not performed before sorting. As stated in the workflow (Figure 1a and L 102-111) synaptosomes are immobilized on coverslips and then go through immunofluorescence staining and analysis. We now explicated this better in the results part of Figure 2 (L 171-174). Part of the confusion may come from the dot plot representation we use to gate our populations of synaptosomes with both staining. These are not from the FACS but derived from the quantification of microscopy images as shown in the workflow. The preparation of samples on coverslips is detailed in the methods (L 743-746) as well as immunofluorescence procedures (L 778-786).*

Finally, sorting is done with endogenous EGFP and anti-EGFP labelling is done to ensure brighter and more stable signal for immunofluorescence analysis. The whole protocol is also published for VGLUT1-FASS in Luquet et al., 2017 (PMID: 27943188; Ref #21 in the text). We hope these clarifications answer the reviewer's point.

Minor points:

- The miniaturization of the centrifugation methods used to isolate synaptic and subsynaptic elements could be better explained in the methods, as these might be of use to other groups.

The full detailed protocol of our miniaturized fractionation was published in De-Smedt-Peyrusse, et al., 2018 (doi:10.1007/978-1-4939-8739-9_5) and cited in the text as ref #25. A detailed protocol of the DA-FASS procedure with the miniaturized synaptosome preparation is going to be published in an invited book chapter to be submitted in April this year.

- It is my understanding that when the FASS sorted material is filtrated using 0,1um filters the material attached to the filter is what is later used for WB or MS; not the material that passes through the filter. It should be clarified how this is done.

We have implemented more details of the filtration, recovery and titration procedure (L 746-747 / 754-755). A detailed protocol of the DA-FASS procedure with the whole protein handling procedure is going to be published in an invited book chapter to be submitted in April this year.

- It might be interesting to discuss why in LP2 (Fig 1C) there is some Th while no EGFP. There might be some TH associated to SVs or other membranous structures.

Indeed a small fraction of Th seem associated to light membranes of LP2. A sentence has been added in the results section (L 124-126).

- In Fig 1A3 (or the figure legend) it would help to clarify that B correspond with the synaptosomal fraction (B=SYN in Fig1C)

B has been systematically replaced by the abbreviation SYN in all panels and in the text.

- Fig1AE in x axis legends are not well displayed

Corrected.

- Lane 136 / Lane 137: I think the text 'a significant proportion' does not refer to any statistical analysis. The term 'significant' is normally used in the context of a statistical analysis. More importantly in several instances throughout

the text vague language is used (as in here). This should be avoided, providing numeric values whenever possible. *We have removed these terms. As our EM analysis is only qualitative, we did not include quantitative data.*

- Lane 146. It is unclear in what sense the word 'specific' is used in this context. Why specific? Consider re-writing in a more precise style.

In line with major point #1, this sentence has been replaced by: "Beyond displaying axo-axonic or axo-spinous synapses, our fractionation isolate multipartite bound synaptic elements that we name "hub synapses".

- Could help the general audience to better define what are unsorted singlets.

We removed most reference to "unsorted" or "sorted samples", "singlet ...". We now modified our nomenclature to ease reading. We considered the 3 levels that must be defined in our workflow. These are:

1) The samples:

Conventional synaptosomes are called "SYN" (and come from three different preparations: DAT-Cre-EGFP or DAT-cre-mNeonGreen, or VGLUT1^{venus}).

Sorted samples are called "DA-FASS" or "VGLUT1-FASS".

2) The flow cytometry or Immunofluorescence gates:

Gates are systematically named based on the markers quantified. for example "EGFP+" or "mNeonGreen+".

3) Individual particles or populations of particles:

Particles are named according to the markers labelled. for example "EGFP+/Th+".

These are now displayed on Figure 1a to guide better the reader.

We applied this new nomenclature in all figures and texts.

- Lanes 158/159 and 162, please add reference to figures. Data in lane 162 is referred to Fig S2, no?

Corrected

- Lane 171: Clarify how this 55% percent is calculated. I understand it is from $32/(26+32) \times 100$.

This is correct. We have added the corresponding calculations when necessary.

- L182: 'nearly all': unprecise, please try to add numeric values to these statements when available.

Removed. we have added quantifications and statistics throughout the text. Hence, we left the sentence like this: "Altogether our data supports that dopaminergic synaptosomes carry a post-synaptic element equipped with cognate receptors."

- Typo: L187: 'spectrometric' should be 'spectrometry'

Corrected

- Would be interesting to discuss why dopamine receptors are not found enriched in the MS analysis. Actually it is my understanding that only D1 is actually identified among the 26700+ proteins found by MS.

Now discussed at L 551-556

- L237: shouldn't CIN cells be in figure 4A and S3.

CIN are represented as interneuron Chat in Dropviz database. The text has been modified (L 270-271).

- L263: SynCAM2 is used together with DAT, while all other proteins are analysed with TH, why? Same thing in L331: VACHT is analysed with TH and the other transporters with EGFP, why?

This is due to the combinations of primary-secondary antibodies compatible for these double labeling immunofluorescences. When it was not possible to use the anti EGFP in combination with another marker, we replaced with the best available combination of markers (either DAT or Th).

- L709: typo, 'spots' should be 'bands'

We replaced "spots" by "samples".

- Adding a table in supplementary with all details regarding the large number of antibodies used could be quite useful.

Thank you for the suggestion it is now done in supplementary Table 1

Reviewer #3 (Remarks to the Author):

The striatum is a brain region that is involved in cognitive, motor and emotional processes. Medium spiny neurons, the principal neurons of the striatum, receive inputs from glutamatergic, GABA-ergic, but also dopaminergic

neurons. Despite the known modulatory effects of dopamine on glutamatergic and GABA-ergic synapses, as well as the physiological and medical importance of the dopaminergic pathway, there is still no consensus regarding the relative importance of synaptic and extrasynaptic dopamine release in the striatum,

The authors developed a cell fractionation and fluorescence sorting procedures to obtain striatal synaptosomal fraction where dopaminergic inputs are fluorescently labeled. This allowed them to characterize the molecular composition of the synapse types present in the striatum by immuno-labeling and mass spectrometry, as well as the relative localization of synaptic components. Combining this data, they provide evidence that glutamatergic and GABA-ergic synapses associate with dopaminergic presynaptic terminals to form a structure termed hub synapses. While dopaminergic afferents localized in the vicinity or making synaptic contacts with glutamatergic synapse have been observed by electron microscopy before, the characterization of these structures would make this manuscript suitable for Nature Communications. However the manuscript does not seem to be completely finished, there are many (mostly small) mistakes and the data is not always presented in a way that clearly leads to the conclusions. Here are the specific concerns:

1. Parts of the manuscript are difficult to find. Specifically, it should be clear which files contain supplementary tables. Currently, these files have cryptic names, one can guess which ones are Tables S3-S5 from their content. There are no references for Tables S1 and S2, but there are two additional tables that are not labeled.

2. This is a complex manuscript that uses lot of abbreviations. In its current form, the terminology used makes some parts of the manuscript difficult to read. Therefore, I suggest that the terminology is improved, the abbreviations should be consistent and it is not advisable to use different terms that have the same meaning. For example:

We thank the reviewer for this point. It is obvious that our story in its previous version suffered from the accumulation of different layers of corrections and nomenclatures due in part to the novelty of the objects and methodologies we are working on.

We removed most reference to “unsorted” or “sorted samples”, “singlet ...”. We now modified our nomenclature to ease reading. We considered the 3 levels that must be defined in our workflow. These are:

1) The samples:

Conventional synaptosomes are called “SYN” (and come from three different preparations: DAT-Cre-EGFP or DAT-cre-mNeonGreen, or VGLUT1^{venus}).

Sorted samples are called “DA-FASS” or “VGLUT1-FASS”.

2) The flow cytometry or Immunofluorescence gates:

Gates are systematically named based on the markers quantified. for example “EGFP⁺” or “mNeonGreen⁺”.

3) Individual particles or populations of particles:

Particles are named according to the markers labelled. for example “EGFP⁺/Th⁺”.

These are now displayed on Figure 1A to guide better the reader.

We applied this new nomenclature in all figures and texts.

(a) Please define “effector synapses”. It might be even better to use “glutamatergic and GABA-ergic synapses” instead.

We now systematically replaced “effector” by “glutamatergic or GABAergic”.

(b) (Lines 124-6): Please explain differences between “DAT-EGFP”, “singlet EGFP” and “FASS EGFP”, a simple scheme might help.

*As stated above, the whole nomenclature is now unified. The sentence is now : “Our gating strategy was adapted from previous work ²⁷ to avoid sorting aggregated particles, i.e. particles with high forward scatter (FSC) and side scatter (SSC) values and sort specifically singlets, particles with FSC values around 0 (Supplementary Fig. 1a-c). Among singlets, EGFP⁺ events are specifically detected by setting a fluorescence threshold from the auto fluorescence of synaptosomes from non-injected mice (Supplementary Fig. 1b). Synaptosomes from DAT-Cre-EGFP mice (SYN) contained on average 3.86 ± 0.53 % EGFP⁺ synaptosomes (N = 9 sorts; Figure 1e,f). Upon reanalysis of the DA-FASS sample in the cell sorter, EGFP⁺ events represented around 50 % of the total (48.9 ± 2.3 %, N = 8 sorts; Figure 1f) and EGFP⁻ synaptosomes were concomitantly depleted (SYN: EGFP⁻ Singlets = 66.1 ± 4 %, N = 9 sorts; DA-FASS: EGFP⁻ Singlets = 30.9 ± 2.8 %, SYN - DA-FASS: Šidák’s multiple comparison ****p < 0.0001 N = 8 sorts; Figure 1f)”*

We hope that these changes make our report much more readily accessible?

(c) (Figure 1F): The use of “singlets” is inconsistent and confusing. The two bars on the left (a dark and the light green) are labeled “singlets” below the x-axis, however the three dark green bars are labeled “singlets” in the legend. Also, I suppose EGFP⁻ and EGFP⁺ bars are also singlets (as opposed to aggregates), which makes all of the bars singlets, regardless of the labeling. Please clarify what the bars mean.

Indeed, this was very confusing. As stated above, the whole nomenclature is now unified. The figure has been changed accordingly.

(d) Another example is “unsorted singlets” as opposed to “FASS-sorted synaptosomes” (line 153), even though the latter are probably also singlets. The same for “Singlets” and “DA-FASS” on figures.

Corrected according to the new nomenclature.

(e) Do DAT-EGFP, DA-FASS and EGFP+ FASS mean the same? If yes, please use only one.

Corrected according to the new nomenclature.

(f) DAT+/EGFP-, EGFP+/D1R- and EGFP+/D1R+ are examples of clear nomenclature. I suggest using this form for all other cases, for example DAT+/EGFP+ instead of DAT-EGFP

Corrected according to the suggestion of the reviewer on all figures and texts.

(g) There is a change from “DAT-EGFP labelled synaptosomes” to “DAT-EGFP labelled varicosities” (lines 166, 169). Please state whether synaptosomes and varicosities are the same.

Corrected according to the new nomenclature. Technically, the part of the synaptosome that is labelled is the varicosity. Now all are named synaptosomes. (L 171-173)

(h) Labeling like “Th+/EGFP-” and “Th-/EGFP+” instead of “Th+” and “EGFP+” would make bar charts easier to understand (Figure 2C, ...)

Corrected according to the suggestion of the reviewer on all figures and texts.

3. Shouldn't the ratio of EGFP+ to EGFP- (the upper two quadrants to the lower two) be the same on Fig 2 A, E and I because the sample is the same. This ratio is much higher on panel A than on E and I.

This is unfortunately not the case because only labelled particles are taken into account. We do not have access to all particles for counting but only those stained by at least one of our antibodies. Hence even though samples are the same, the number of Th+ particles is different from the number of D1R+ particles for instance. Th molecules are mostly present at EGFP+ particles but D1R molecules seem to populate many particles, coming from the D1R+ SPNs, devoid of EGFP molecules. Only a minority of D1R+ particles are at EGFP+ synaptosomes.

Ideally, we would like to have access to a monitoring of the whole population of particles but to date we didn't find a technical solution allowing both monitoring all particles and performing double immunofluorescence staining.

4. Regarding the statistical analysis:

(a) At several places, a change in percentage of dots in a quadrant is reported in the manuscript, but no statistical significance is given (line 177, 293, ...).

Corrected.

In a previous editorial process reviewers asked to remove all statistics from the main text and group them in a separate file ...

To follow the guidelines of Nature Communications we now inserted statistics in the main text. We also kept a separate file (Supplemental Table 2) with all details of the statistical tests performed.

(b) It is stated that two-way mixed design ANOVA was used and the results are shown in one of the unlabeled files (Figures like 2C, 2G, ...). However it is not clear what are the ANOVA factors (fixed treatment and random effect), because the sorting, EGFP and another factor (Th, D1R, ...) are present.

This has been corrected. We used a Fixed model two-way ANOVA with two factors. Condition represents the SYN and FASS condition and Immunolabelling the 3 different types of particles immunolabelling, and not a mixed design ANOVA. This error stems from a previous manuscript version.

(c) Whenever the statistical significance is stated, what is shown to be different

*Corrected. We now provide all details in the figure legend, e.g. Figure 2c “All data are mean \pm SEM and pulled from 2 to 3 independent sorts. Statistical significance was analysed using Two-way ANOVA; c EGFP/Th: Interaction $F_{2,96} = 65.04$ **** $p < 0.0001$, Condition $F_{1,96} = 0.03371$ $p = 0.8547$, Immunolabelling $F_{2,96} = 510.3$ **** $p < 0.0001$ with Šidák's multiple comparisons test) Additionally, the p-value of Šidák's multiple comparisons test for each immunolabelling between the two sorting conditions is now indicated in all the figures using a line for each comparison and in the main text for relevant comparisons. See figure 2 for examples.*

5. It is confusing to find numbers in the text that are not shown on the corresponding figures. For example, I am aware that “27%” (line 323) is correct because it comes from “20% and “54%” (Figure 5F). However, making these obvious would help the reader

All occurrence corrected. First, we checked all numbers and found a few mistakes accumulated in a previous round of reviews. We apologize for these. Second, we now provide all details regarding the calculation of this number. In

this example 26% comes from the calculation of VIAAT+/EGFP+ (19%) particles among all EGFP+ particles (54%). $(19/(19+54) \times 100 = 26\%$; L381-382).

6. Given the proteins investigated in this section, it would be more precise to use “dopaminergic presynapses” and “all the major presynaptic partners” (lines 335-6). Similar issue on line 351
Corrected (Now L400).

7. How are “co-localized”, “apposed” and “rather distant” defined (lines 367 and 369)? Quantification is needed for these statements in order to justify the model (lines 372-5). Does “either D1 or D2” come from the data presented here or elsewhere? It is not clear how is the sentence on line 365 related to the model.
We have reduced the use of imprecise terms. The whole synaptic model has been enriched with the calculation of distances of Stx4. We have added the STED imaging for Stx4. Figure 6 is modified accordingly. The text has been modified to precisely define our current model from data gathered in Figure 2, 4, 5, and 6 (L 427-434). Regarding the association of dopaminergic synaptosomes to D1R+ or D2R+ membranes, we calculated the percentages from our data. We explicit this in the corresponding section of results associated to Figure 2 (L189-205).

8. There are several issues regarding the detection of Bassoon:

(a) Please state the p-value (lines 390, 397)

Done.

(b) The qualification “close to the active zone” is not supported by Figure S6A (line 393)

Removed.

(c) It is not clear why Th-/VGLUT1+ is compared with Th+/VGLUT1- and which conclusions can be made from this (lines 392-4)

We measured Th-/VGLUT1+ as a control for the Bassoon signal on our preparations, at Th+/VGLUT1- to evaluate the abundance of Bassoon at dopaminergic varicosities. We didn't mean to compare both values per se.

(d) Two way ANOVA analysis of this data might determine the interaction between VGLUT and Th

As pointed in (c), there is no rationale to make this comparison.

(e) STED data need to be statistically analyzed in order to justify the conclusions (lines 399-402). The possibility that the higher Bassoon signal in Th+/VGLUT1+ compared to Th-/VGLUT1+ is simply due to the contribution from those Th presynapses that contain Bassoon.

Based on the reviewer's comments, we have now reworked the whole part on Bassoon. We implemented additional STED imaging experiments and measure Bassoon clusters in Th+ regions of interest (ROI) and VGLUT1+ regions of interest from 166 synaptosomes from 3 independent preparations (Figure 7e; Supplemental Fig 6d). We measured a 33% increase of Bassoon when VGLUT1 synapses are engaged in a hub. Yet, only 40% of associated Th+ varicosities displayed a detectable Bassoon signal in these hubs. Furthermore, the intensity of Bassoon within Th+ ROI, represented less than 25% of the average Bassoon intensity at VGLUT1+ ROI. From this analysis we conclude that Bassoon increase at hub synapses cannot be simply attributed to the added Bassoon present at the dopaminergic presynapse. We also added new data to monitor Rim1 (Supplemental Fig 6c). We added Rim1 because of the low representation of Bassoon at dopaminergic synaptosomes and the article from Pascal Kaeser that shows the requirement for Rim1 presence to release Dopamine (Liu et al, 2018 PMID: 29398114; ref #13 in the text). We detected Rim1 at 71% of Th+ presynapses regardless of the engagement in hubs with an average intensity lower than for Rim1 in VGLUT1+ presynapses.

9. Statistical significance should be stated for results on lines 406-417)

All statistics have now been added throughout the text.

10. Is it possible that dopaminergic presynapses do not make synaptic contacts to glutamatergic (or GABA-ergic) postsynapses as proposed for hub-synapses, but that dopaminergic and glutamatergic (and GABA-ergic) synapses do not share the postsynapse and are kept together by the direct association of the presynaptic terminals, like on Figure 1J?

It is indeed possible that some Th+ presynapses impinge on glutamatergic presynapses, to illustrate this we show the 2 possibilities with question marks on our model in Figure 7j. It has been documented previously as quoted in ref #5 by Moss and Bolam. In such occurrence, we can speculate that synaptic transmission may occur at the contact area between the 2 boutons.

11. It is not clear on which basis this assumption is made (line 504)

This was modified with the exact number of 41% and the corresponding calculation (L 613-614).

12. As noted above, the statements on lines 522-524 are not justified

The text has been modified according to all reviewers' comments. Reference to figure 7 was added. The correlative nature of our observation is now stressed, causality is not suggested (L 634-637).

13. The method for measuring distance between two labeled dots should be specified in more detail:

(a) Is the source code available?

In the present manuscript we worked to provide a unified algorithm with graphical user interface in imageJ/fiji for easy implementation by other users. This software's source code available at : https://github.com/fabricecordelieres/IJ-Toolset_SynaptosomesMacro for the correlation of synaptosomes' labelling containing distances measurements (L 870) and at <https://github.com/flevent/RandomizerColocalization> for the association analysis plugin (L 905).

(b) What features are shown by the "binary representation"?

This part was present in the previous association analysis plugin which was removed from the unified version. The "binary representation" represented segmented particles in grey and not segmented space in black.

(c) Please explain how was the wavelet filtering used

In the previous version of the manuscript, we have used the à-trous wavelet segmentation method, implemented in SpineJ¹, to segment immuno-labels as individual objects. Briefly, à-trous wavelet computes a series of multi-scale wavelet coefficients by iterative convolutions of increasing kernels. These kernels are derived from a low-pass filter and stretched by adding zeros with each new wavelet coefficient level. A-trous wavelets exhibit several key features: (i) the noise variance $\hat{\sigma}_\epsilon$ can be robustly estimated²; (ii) the size of filtered objects is directly related to the wavelet scale, allowing segmenting structures of similar size by thresholding a given wavelet sub-band; (iii) wavelets are not sensitive to absolute image intensities, making it possible to quantify and compare different images. Consequently, we only used the second wavelet sub-band and used a threshold of $30 \hat{\sigma}_\epsilon$ for all the images analyzed in this study. Results of this filtering were two binary images (one for each staining) with immuno-labels being identified as individual objects.

1. Levet, F., Tønnesen, J., Nägerl, U.V. & Sibarita, J.B. SpineJ: A software tool for quantitative analysis of nanoscale spine morphology. *Methods* 0–1 (2020). DOI:10.1016/j.ymeth.2020.01.020.
2. Donoho, D.L. & Johnstone, I.M. Adapting to unknown smoothness via wavelet shrinkage. *J. Am. Stat. Assoc.* **90**, 1200–1224 (1995). DOI:10.1080/01621459.1995.10476626.

However, in order to use the same segmentation procedure between all analysis of the image set, we now replaced the wavelet filtering by the segmentation used for the synaptosomes immunofluorescence intensity analysis described in the methods (L 861-868). The full procedure is now detailed in the methods (L 857-909) and at https://github.com/fabricecordelieres/IJ-Toolset_SynaptosomesMacro.

Briefly, in the new unified version, segmentation is performed using the following steps: a maximum intensity projection is performed; the projection is subjected to gaussian blurring; the image is then median filtered and a search for local maxima is performed to identify objects/particles; for each retrieved maximum imageJ "magic wand" tool is activated on the spot, generating a ROI surrounding each object/particle. After this step a square ROI is centered over the picked point and is added to the ROI Manager; all ROI centers' coordinates are stored. Having the two centers' coordinates (one per channel), the distance between both is computed, calibrated (according to the provided calibration, 0.162 or 0.103 microns/pixel) and logged to the results table.

(d) Are "objects" and "particles" the same?

Objects and particles are indeed the same

(e) What was the size of analyzed images? Like Figure 2D?

Figure 2D are STED images. Distance measurements were performed on epifluorescence images. Their size is therefore (2048x2048 pixels). In the present version, all distances were measured with the unified macro/plugin. It allowed us to focus the distances measurements on the exact same set of particles identified for intensity measurements, giving us a more precise dataset. New distances are now present in the text (L427-434). The randomization was performed on every segmented particle.

(g) Was the randomization done over the whole image or a portion of it?

As written above the randomization is done over all detected particles spanning the whole image.

(h) Isn't the probability of having a random particle at $< 2 \mu\text{m}$ to the fixed particle simply the fraction of the image area that is $< 2 \mu\text{m}$ to the fixed particle?

This part was not clear enough in the methods, we adjusted the text accordingly (L 908-919). After retrieving previously stored particles centroids coordinates. For each color channel, we fixed the position of its particle while randomizing all the ones of the other channel using 10 000 Monte-Carlo simulations. Since there is no underlying structure, the probability of having a not fixed particle at a certain position is identical for the whole image space.

14. Considering that the extracellular region of SynCAM2 was labeled, why is the SynCAM2 label relatively far from the DAT label (Figure 6G)?

We are surprised by this point. In fact, the center of SynCAM 2 signal is seen on average at a distance of 285nm from the center of the dopaminergic presynapse labelled by DAT. This distance is fairly small compared to other markers tested. It is possible that DAT molecules are present more evenly around the varicosity, while SynCAM2 maybe concentrated at adhesion sites. However, this is pure speculation at this stage and labeling with a putative post-synaptic partner and other controls would be needed to test this hypothesis.

15. Synaptosomal preparation:

(a) The synaptosomal preparation used is termed "the classical sucrose fractionation" (line 103), while the ficoll preparation is described in the Methods.

Indeed this is a mistake. We corrected the text now (L 105). In fact, when the project started we were using sucrose gradients, because we have slightly better results with ficoll in terms of fluorescence signal in the cell sorter and for functional assays (not implemented in the current work) we then moved to ficoll for most of the experiments of the project. While both types of gradients may be used almost indifferently for most data we preferred to propose the ficoll version in the methods. We will submit a detailed protocol of DA-FASS in April this year for publication that will detail all our state-of-the-art procedures.

(b) How was the number of synaptosomes determined (line105)

The FACS counts events sorted. The text has been modified accordingly (L 127-128).

16. Please explain "gelatinized coverslips (line 633)

Details have been added on this procedure (L 743-746). The detailed procedure is published in ref #21.

17. (Line 119-120): "Soluble proteins" is not clear. What is "S2"? It cannot be said "most of the soluble protein ..." when only two proteins are checked (EGTA and Th).

With this experiment we wanted to test whether our fractionation may be biased if much of the content of dopaminergic presynapses was actually leaking from broken presynapses at the homogenization step. This point was raised by a reviewer in a previous editorial round. We therefore applied a complete classical fractionation of the tissue that is an extended fractionation compared to the synaptosome preparation. We probed these fractions for 2 soluble proteins of DAT-cre-EGFP dopaminergic axons, The Tyrosine hydroxylase and EGFP. It is standard in the field to use one or two markers of a given compartment to probe the behavior of the proteins of this compartment in the fractionation. Because we find the strongest enrichment of our 2 soluble markers in the SYN, LS1 and LS2 fractions, we are confident that most of the content of dopaminergic presynapses is preserved from leakage at the homogenization step and ends up in the synaptosomes prior to FASS.

S2 is the fraction containing soluble proteins released from cell soma during homogenization and super light membranes from cell soma (endosomes, lysosomes, ER).

The text has been corrected to answer the reviewers' point (L 114-126).

18. (Line 132): Munc18 appears more than "slightly reduced" on Figure 1G

Corrected (L 151-152).

19. Please quantify the statements "significant proportion" (line 136) and "another example" (line 139)

These terms have been removed to stay with a purely qualitative description of our sample in electron microscopy. Our data are not suited for a proper quantitative analysis and we preferred to stick to a qualitative description at this stage. We are working on a more advanced electron microscopy approach to explore our samples in a quantitative manner but this is too preliminary to be implemented in this first article on dopamine synapses. (L 153-163)

(a) "Collapse aggregates" is not clear (line 145). Would there be a possibility that cell sorting induces aggregation?

We indeed assumed too fast that this point is clear for the reader. We addressed these issues in previous studies (ref #21 and #26). We have no evidence for aggregation triggered by the cell sorter itself. In supplemental Fig 1 and 5, one can see that upon sorting of singlets, the aggregates representation drops. In supplemental Fig 5hi we show that when sorting aggregates and large particles, upon reanalysis a large fraction of the sample is in the singlet gate. The most parsimonious explanation is the collapse of aggregates upon flight in the instrument nozzle known to generate shearing forces. Large cellular debris are shown in supplemental Fig 5f. Our conclusions were confirmed by an independent study by Benjamin D Hobson (Hobson et al., 2019 PMID: 31118205; now ref #28). The text has been modified (L 163-166).

20. What criterion was used to determine the intensity separation lines on dot plots (Figure 2A, B, D, F, ...)?
As pointed by the reviewer, gate limits for immunofluorescence analysis are a critical parameter important to define populations of synaptosomes. Distributions of fluorescence intensities in the ROI usually display a first low intensity peak for background and one or several high intensity peaks for positive particles. We identified the range of fluorescence that is between the background peak and the next peak. In this range of intensities, we then observed several ROIs. The gate border was placed at the last intensity values where no clear puncta can be observed in the ROI for channel. This method was used for all immunofluorescence quantifications in all figures even though we only show the dot plots in Figure 2 for a sake of space and simplicity. The text was modified to include this explanation (L 895-900).

21. Please explain where is shown the “5-fold” increase (line 159)

The data concerning number of particles per frame were removed from the current version of our study. We omitted to remove the corresponding sentence for Th staining. Of note the change in particles per frame between SYN and DA-FASS samples is related to the point 3 of the reviewer. We now removed this sentence from the text.

22. Co-localization of what with anti-DAT (line 161)

The sentence was changed for : “Similarly, we found a strong co-localization of EGFP⁺ synaptosomes with anti-DAT (Dopamine Transporter) signal (SYN: EGFP⁺/DAT⁺=14.5%; DA-FASS: EGFP⁺/DAT⁺=47%; N=1 sort; Supplementary Fig.2ab)”. (L 183-184).

23. The argument on lines 162-167 is confusing.

This part was modified (L 182-187).

24. No justification is given for calling (EGFP-) D1R extra-synaptic (line 170)

Removed and modified with the new unified nomenclature (L 190-193).

25. Please quantify “closely or more distantly” (line 179)

Corrected, we removed a reference to the distance as we do not have the data to quantify this on a population scale in STED (L 206).

26. Please specify “very high” (line 288)

we now provide numeric values (L 333-335).

27. It is not clear where do the statements about apposition (lines 294 and 301) come from

Sentences have been modified (L 340-342 and 349-350).

28. Please be more specific (lines 207, 306)

We implemented a sentence with numeric values (L 238-240).

29. Considering that immuno-fluorescence signal centers are found for each protein separately, wouldn't “center of mass” be more appropriate than “barycenter” (spelled wrong) (line 359)

Corrected to “center” (L 422).

30. Figure 5D is not related to Th (line 380)

Indeed, it refers to dopaminergic synaptosomes probed with EGFP. We modified the text accordingly (L 453).

31. Rather “decreased” than “maintained” (line 384)

We do measure over the triplicate that Th is maintained through VGLUT1-FASS, while VGLUT1^{venus} is increased, The data is now representing this better and statistical analysis is provided (Fig 7c, L 457-459).

32. “Significant” would be more precise than “strong” (line 427)

Corrected (L 524).

33. References 4 and 5 are cited as supporting two opposite conclusions (line 489-91)

Corrected, references were mistakenly inserted in the second citation (L 599).

34. Figure 7J is not cited.

We included a reference to the model proposed on Figure 7j (L 521-522).

35. Please state the pixel size wherever distances are specified in pixels (lines 777, 818)

Corrected (L 887-890 / 939-940).

36. Which “.CSV file” (line 791)?

File name is added (L 893).

37. Figure 1C is not cited in the correct context (line 795)

Reference to Figure 1C was removed (L 902).

38. Symbols “b” and “m” are not specified for Fig 1L (line 1154)

Added in the figure legend (L 1278-1279).

39. Please state N (line 1252). Also, where do “55%” and “45%” come from (line 1259)?

Statistics are added. An explanation for D1R/D2R percentages is now added (L 1383-1386).

40. Where do “60%” and “40%” on Figure 7J come from?

An explanation is now given (L 1400-1402 and 1420).

41. There are several cases of spelling mistakes and inappropriate word use. For example: “aimed” (line 85), “decisively” (line 311), “challenge” (line 337), “co-sedimentation” (line 339), words missing (line 349), “each markers” (line 359), “reports for” (line 387), “harbour a cluster of Bassoon” (line 392), “until now on the other hand” (line 449), “engage into” (line 511), “are clear to electrons” line (512-3), “plan” (line 1154), “VGLUT2 thalamo” instead of “VGLUT1 thalamo” (line 1343).

All points were corrected and the new text was submitted to a native English speaker for corrections.

REVIEWERS' COMMENTS

Reviewer #1 (Remarks to the Author):

This manuscript describes a methodological approach in which fluorescence-activated particle sorting was used to enrich synaptosomes (termed FASS) from dopaminergic terminals in striatum, followed by mass spectrometry to determine protein identity and abundance. Extensive, follow-up validation using immunological probes and high-resolution imaging was then carried out to characterize the proteome organization of dopaminergic synapses formed on striatal spiny neurons.

General comments:

The original version of the manuscript described work that was carefully done and well carried out, with adequate secondary validation and quantitation. The work was considered to be a useful technical report for researchers studying striatal biology. For this reviewer, there were two main criticisms, the first being a concern over the quality of the writing (also shared by other reviewers). In this regard, the manuscript is improved, though the Discussion is overly long and would be improved by shortening its more repetitive parts. The grammar and sentence structure in the Methods could also be improved. The second criticism was related to the fact that the work is inherently descriptive with no biological manipulations. This led the authors in a number of places to infer causality, when only correlation is shown. To a large extent the text is improved, but the title of the section on L444 – “Dopaminergic innervation strengthens VGLUT1 excitatory cortico-striatal synapses” infers more than is actually shown.

Specific comments:

1. This is a question I should have raised before, but is there any additional quantitative or qualitative data from the EM summarized in Figure 1h-l, related to the frequency/proportion of the different structures shown?
2. L225 – Fig 3a/g – I count 61 proteins depleted – not 63?
3. L263 – not all marked with # in Fig 3g?
4. L524-526 – sentence seems like a circular argument.
5. L576 – “the integrity [of the?] dopaminergic system”
6. L634 – what does “protein content” mean?
7. L832 – “fluorescent”
8. L833 - Precursor Detector “node” - mode?

Reviewer #2 (Remarks to the Author):

In my opinion the authors have made a very thorough work responding comments from all reviewers. Including mines.

The manuscript has been extensively re-written and is now easier to follow. Some new data has also been added.

The issue raised on the statistics of MS-data seems to have been clarified. It wasn't properly described in the previous version.

Overall, this is an interesting new addition to the field that deserves publication in a journal with a wide audience such as Nat Comms.

I still would recommend the authors to look for a term which is more precise than 'hub synapses'. I've seen in the manuscript the use of 'Dopaminergic Hub synapses'. This, or a similar, more constrained term could be more effective at conveying their message. I do not think the nomenclature of 'Hub Synapses' will be accepted and generally used, as it does not bare any reference to the dopaminergic system itself. Yet this is a matter of personal opinion.

I would also recommend the authors to more openly present/discuss the first works describing the existence of dopaminergic buttons contacting other synapses. I also still belief that the new findings have more to do with the fact that most DA synapses are directly contacting other synapses, rather than forming synapses of 'their own'.

Reviewer #3 (Remarks to the Author):

The authors performed additional experiments and addressed addressed all points raised by this reviewer. I recommend the publication of this manuscript without further revision.

Reviewer #1 (Remarks to the Author):

This manuscript describes a methodological approach in which fluorescence-activated particle sorting was used to enrich synaptosomes (termed FASS) from dopaminergic terminals in striatum, followed by mass spectrometry to determine protein identity and abundance. Extensive, follow-up validation using immunological probes and high-resolution imaging was then carried out to characterize the proteome organization of dopaminergic synapses formed on striatal spiny neurons.

General comments:

The original version of the manuscript described work that was carefully done and well carried out, with adequate secondary validation and quantitation. The work was considered to be a useful technical report for researchers studying striatal biology. For this reviewer, there were two main criticisms, the first being a concern over the quality of the writing (also shared by other reviewers). In this regard, the manuscript is improved, though the Discussion is overly long and would be improved by shortening its more repetitive parts. The grammar and sentence structure in the Methods could also be improved.

We now shortened and simplified the discussion. We agree with the referee that this was necessary to further improve the article. We also corrected the methods accordingly.

The second criticism was related to the fact that the work is inherently descriptive with no biological manipulations. This led the authors in a number of places to infer causality, when only correlation is shown. To a large extent the text is improved, but the title of the section on L444 – “Dopaminergic innervation strengthens VGLUT1 excitatory cortico-striatal synapses” infers more than is actually shown.

This was now corrected to : “Comparison of VGLUT1 excitatory cortico-striatal hub versus regular synapse.” We carefully ensured no overinterpretation of our findings remained throughout the text.

Specific comments:

1. This is a question I should have raised before, but is there any additional quantitative or qualitative data from the EM summarized in Figure 1h-l, related to the frequency/proportion of the different structures shown?

To address previous criticism of another referee we removed any unprecise notion of frequency from the EM part of the article. In fact, our data set is too small (N=1) to allow for quantitative analysis. Yet we did observe multipartite elements frequently during the observation of the grids. We are now engaged into a thorough EM analysis of dopamine hub synapses using a Cryo-CLEM workflow, though this goes out of the scope of the current study.

2. L225 – Fig 3a/g – I count 61 proteins depleted – not 63?

We are grateful to the referee for pointing this mistake. It is indeed 63 proteins. 2 proteins escaped during the making of figure 3g. Now corrected.

3. L263 – not all marked with # in Fig 3g?

We are grateful to the referee for pointing this mistake. Now corrected.

4. L524-526 – sentence seems like a circular argument.

The sentence is now shortened to : “This observation strengthens the notion that dopamine hub synapses represent a mechanically resilient functional structure.”

5. L576 – “the integrity [of the?] dopaminergic system”

corrected

6. L634 – what does “protein content” mean?

Changed to “synaptic markers”

7. L832 – “fluorescent”

corrected

8. L833 - Precursor Detector “node” - mode?

It is indeed an optional “node” in the workflow of Proteome discoverer 2.5.

Reviewer #2 (Remarks to the Author):

In my opinion the authors have made a very thorough work responding comments from all reviewers. Including mines.

The manuscript has been extensively re-written and is now easier to follow. Some new data has also been added.

The issue raised on the statistics of MS-data seems to have been clarified. It wasn't properly described in the previous version.

Overall, this is an interesting new addition to the field that deserves publication in a journal with a wide audience such as Nat Comms.

I still would recommend the authors to look for a term which is more precise than 'hub synapses'. I've seen in the manuscript the use of 'Dopaminergic Hub synapses'. This, or a similar, more constrained term could be more effective at conveying their message. I do not think the nomenclature of 'Hub Synapses' will be accepted and generally used, as it does not bare any reference to the dopaminergic system itself. Yet this is a matter of personal opinion.

We propose to restrict the term to "dopamine hub synapse". We also thought of "Dopamine tripartite synapse" but we feel that this would be too confusing because of the use of "tripartite" to refer to an association with glia in the field of synapse biology. We modified the naming everywhere except when the dopamine nature of the hub synapses is mentioned in the sentence and in the legend of the EM pictures as the dopaminergic nature of the observed hub synapses cannot be certified. We also modified the title to: "A synptomic analysis reveals dopamine hub synapses in the mouse striatum."

I would also recommend the authors to more openly present/discuss the first works describing the existence of dopaminergic buttons contacting other synapses. I also still belief that the new findings have more to do with the fact that most DA synapses are directly contacting other synapses, rather than forming synapses of 'their own'.

We added one sentence in the introduction about synapse identification by previous laboratories.

Reviewer #3 (Remarks to the Author):

The authors performed additional experiments and addressed addressed all points raised by this reviewer. I recommend the publication of this manuscript without further revision.